# A Multi-domain Benchmark for Machine Unlearning in Classification Tasks

## Abstract

Machine unlearning (MU), the process of removing specific data influences from trained machine learning models, is critical for regulatory compliance (e.g., GDPR's right to be forgotten) and for addressing copyright and privacy concerns in large-scale models. While a wide range of methods and metrics have been proposed, systematic evaluations remain fragmented, typically limited in scope by modality, metric coverage, or the number of methods considered. In this work, we present the most comprehensive MU benchmark to date, evaluating 12 unlearning methods on 8 datasets and models across four modalities (images, text, tabular data, and graphs) by assessing the three key aspects of an unlearning outcome: *utility* – the overall performance of the model after unlearning – *efficacy* – how well the data is forgotten – and *efficiency* – the computational cost of unlearning. We also introduce LUMA (Laplacian Unlearning Multidimensional Assessment), a unified metric that consolidates them into a single score. Unlike prior metrics, LUMA can flexibly incorporate multiple measures within each dimension (e.g., F1 over test and forget set for utility, UMIA for efficacy, runtime and GPU memory for efficiency), enabling more accurate and extensible comparisons. Our code is reproducible and extensible to serve as a benchmark for MU research.

## 1 Introduction

The growing inclusion of machine learning (ML) models across various industries has raised concerns about using potentially sensitive data in model training (Grynbaum & Mac, 2023). In response, regulations such as the AI Act and the General Data Protection Regulation (GDPR) established the *right to be forgotten*, mandating that an individual's data must be removed upon request (Mantelero, 2013). In such cases, the model's owner must release a new version of the model itself, specifically excluding those targeted samples from the training set. However, retraining ML models from scratch upon every request is often too expensive in terms of time, money, and environmental costs (Crawford, 2022). *Machine unlearning* (MU) offers a promising alternative: instead of (re)training the model from scratch, MU aims to efficiently remove the influence of specific data points from an already trained model (Xu et al., 2024; Le Quy et al., 2022).

Despite its recent emergence, the literature on MU has been growing at a massive rate, outpacing benchmarking and surveying efforts. Specifically, while methods are often presented as generally applicable (Chundawat et al., 2023; Foster et al., 2024), most benchmarks focus on a single domain, such as images (see Section 2), which limits the ability to assess their generalizability. The Tabular and Graph domains remain largely unexplored: to the best of our knowledge, ours is the first MU benchmark covering these two domains. In contrast, the textual domain has received more attention; however, existing benchmarks target the text generation task (Maini et al., 2024; Shi et al., 2024) rather than the classification one.

In this work, we establish coherence in this fragmented landscape by studying MU methods on diverse domains under a single, unified evaluation benchmark. As illustrated in Figure 1, the proposed unlearning benchmark consists of three steps: (1) selecting datasets from one of the four supported domains, (2) training the appropriate model (between two sizes), and (3) applying unlearning methods directly to the model without requiring domain-specific adjustments.

After unlearning, step (4) is the collection and analysis of the values of the evaluation metrics. As detailed in Hayes et al. (2024), MU consists of three key aspects that must be evaluated jointly:

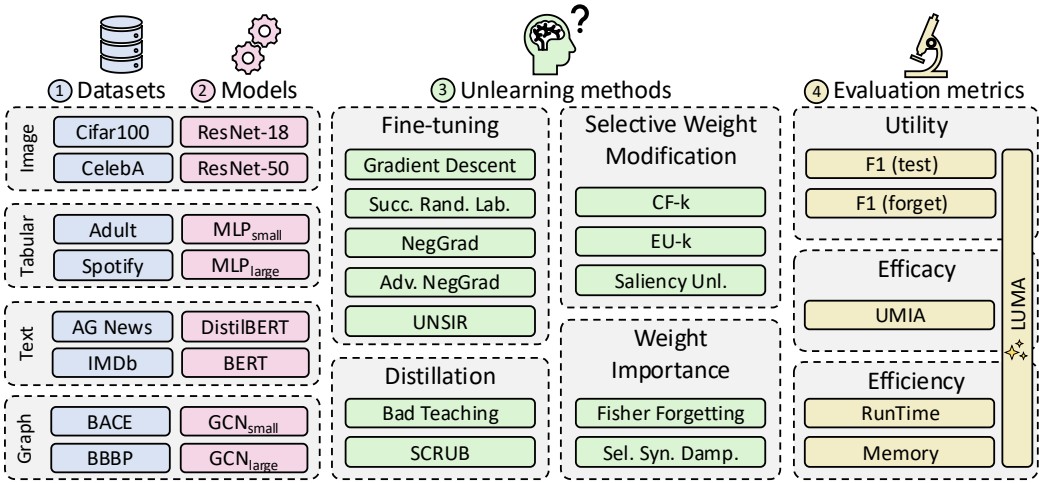

Figure 1: Experimental workflow of our benchmark. We evaluate 8 classification datasets across 4 domains, training 2 models per domain. We test 12 unlearning methods and assess them across 3 evaluation dimensions. We introduce a unified metric, LUMA, to facilitate comparison.

*utility* – how well the model performs after unlearning – *efficacy* – the degree to which the target data is removed – and *efficiency* – the computational cost of unlearning w.r.t. retraining the model from scratch. After presenting each of the three aspects separately, we introduce LUMA, a unified metric designed to bridge the current gap in the literature.

Our contributions are the following: *(i).* We introduce LUMA, a unified metric that jointly captures the three key aspects of MU: utility, efficacy, and efficiency. *(ii).* We validate LUMA on a benchmark of 12 machine unlearning methods on 8 classification datasets spanning four domains (the most comprehensive evaluation of MU methods to date), including the first benchmarks for tabular and graph data. *(iii).* We propose a taxonomy of the 12 benchmarked methods to aid in structuring and understanding the field. *(iv).* The benchmark we provide publicly is reproducible, easily extensible, and intended to serve as a foundation for future research in the MU field of study.

Code is available at this anonymized repository, with results reproducible via `reproduce.sh`.

## 2 RELATED WORK

As an emerging field, machine unlearning has still seen limited systematic evaluation of the extensive research developed in recent years. In addition, most existing benchmarks remain restricted to a single domain. We list these studies in Table 1, providing details on their coverage (in terms of datasets and methods), the domains they include, and the number of aspects of MU they capture with their metrics.

Several works only focus on one domain: Grimes et al. (2024); Choi & Na (2023); Cadet et al. (2024) evaluate and compare unlearning approaches exclusively on image datasets. Similarly, Koudounas et al. (2025) focuses solely on spoken language understanding datasets.

While valuable, no method was tested on multiple domains jointly, an omission that critically limits our understanding of the generalizability of MU methods. Moreover, most benchmarks either do not consider all the evaluation dimensions or are very limited in the number of datasets and methods they employ.

To the best of our knowledge, the benchmark introduced by Cheng & Amiri (2024) remains the only existing effort to evaluate MU methods across multiple domains. However, it lacks a crucial component of MU evaluation: the incorporation of metrics that directly assess the efficacy of the unlearning process. In their absence, the reported results lack rigorous quantification of unlearning effectiveness, undermining the ability to address the core privacy and security goals that MU is

| | Coverage | | Modalities | | | | | | Metrics | | |
|---|---|---|---|---|---|---|---|---|---|---|---|
| | # datasets | # methods | Image | Text | Tabular | Graph | Speech | Video | Utility | Efficacy | Efficiency |
| **Our** | 8 | 12 | ✓ | ✓ | ✓ | ✓ | ✗ | ✗ | ✓ | ✓ | ✓ |
| Cheng & Amiri (2024) | 6 | 5 | ✓ | ✓ | ✗ | ✗ | ✓ | ✓ | ✓ | ✗ | ✓ |
| Choi & Na (2023) | 2 | 7 | ✓ | ✗ | ✗ | ✗ | ✗ | ✗ | ✓ | ✓ | ✗ |
| Cadet et al. (2024) | 5 | 11+7[1] | ✓ | ✗ | ✗ | ✗ | ✗ | ✗ | ✓ | ✓ | ✓ |
| Grimes et al. (2024) | 1 | 6 | ✓ | ✗ | ✗ | ✗ | ✗ | ✗ | ✓ | ✗ | ✗ |
| Koudounas et al. (2025) | 5 | 8 | ✗ | ✗ | ✗ | ✗ | ✓ | ✗ | ✓ | ✓ | ✓ |

Table 1: Comparison of classification benchmarks present in the literature.

meant to achieve (Xu et al., 2024). In addition, as shown in Table 1, their study covers a far narrower set of methods (5 versus our 12), which restricts both the scope and generalizability of their findings.

Table 1 shows that no benchmark in the literature has assessed MU methods in the Tabular and Graph domains. Conversely, our benchmark fills this gap by considering 8 datasets (2 per domain, more than any other benchmark) and 12 methods across four different data domains: Image, Text, Tabular, and Graph. Moreover, we incorporate all key dimensions of MU evaluation (utility, efficacy, and efficiency) within a single framework, introducing LUMA (see Sec. 3.3.1), the first unified multidimensional metric. By providing the most extensive and systematic benchmark to date, we bring coherence to the fragmented landscape of MU in classification tasks and establish an extensible reference point for validating future methods.

## 3 BENCHMARK DESIGN

We formalize the standard MU workflow as follows: a model $M$ is first trained on a dataset $D$. Upon receiving an unlearning request, a subset of samples to be removed is specified as the forget set $D_f$, with the retain set defined as $D_r = D \setminus D_f$. The goal of an unlearning method (Unlearner) is to transform $M$ into an updated model $M'$ that closely approximates a retrained model `Gold` (Gold Model), i.e., the one obtained by retraining from scratch on $D_r$.

We designed our benchmark to faithfully implement this workflow uniformly across datasets. For each method, we perform hyperparameter selection over the learning rate in the range $10^{-6}$ to $10^{-3}$, and additionally tune method-specific parameters where relevant (e.g., the dampening constant in Selective Synaptic Dampening Foster et al. (2024)). All experiments were run three times with different seeds to account for statistical variation.

### 3.1 DATASETS AND MODELS

We evaluate all unlearning methods across four domains – image, text, tabular, and graph – using two publicly available datasets per domain. For domains with established MU benchmarks, we adopt datasets widely used in prior work (Chundawat et al., 2023; Golatkar et al., 2020; Foster et al., 2024; Tarun et al., 2023; Fan et al., 2023; Goel et al., 2022; Cha et al., 2024; Cheng & Amiri, 2024). For domains less explored in this context, we select representative and widely studied datasets (Le Quy et al., 2022; Wu et al., 2018).

The forget set $D_f$ for each dataset is constructed following one of two strategies: (i) selecting training samples containing predefined named entities (e.g., person or organization names) or identity-related attributes **(NE)**, simulating realistic deletion requests; or (ii) sampling 20% of the training set uniformly at random **(SA)** when identity-based filtering is not feasible. In both cases, the forget set size is approximately 20% of the training data, consistent with prior MU literature.

For the **image** domain we selected Cifar-100 (Krizhevsky (2009)) *(SA)* and CelebA (Liu et al. (2015)) *(NE)*. CelebA was used for multilabel classification. For **text** we selected IMDB (Giobergia (2023)) *(NE)* and AG News (Gulli (2005)) *(NE)* for **tabular** data, we selected the datasets of Adult (Becker & Kohavi (1996)) *(SA)* and Spotify Tracks (maharshipandya (2023)) *(SA)*, and for **graphs** we selected BBBP (Sakiyama et al. (2021)) *(SA)* and BACE (Wu et al. (2018)) *(SA)*. More details on these datasets are reported in Section B.1.

---

[1] 7 of the methods reported in this survey were taken from a Machine Unlearning competition and, as such, were not formally peer-reviewed.

For each domain, we adopt model architectures widely used in the MU literature, selecting both smaller and larger variants. In the **image** domain, we use ResNet-18 and ResNet-50 (He et al., 2016), consistent with prior MU studies (Foster et al., 2024; Fan et al., 2023; Golatkar et al., 2020; Tarun et al., 2023). For **text**, we fine-tune DistilBERT (Sanh et al., 2020) and BERT (Devlin et al., 2019) (110M+ parameters), each augmented with a classification head. In the **tabular** domain, we employ two fully connected networks with one and three hidden layers, respectively, each layer containing 100 units. Finally, for **graphs**, we use two GCN-based classifiers: one with a backbone of one GCN layer followed by one dense layer, and another with two layers followed by one dense layer. Full training configurations and hyperparameters are detailed in Appendix B.2.

## 3.2 UNLEARNERS

For this benchmark, we include 12 different state-of-the-art unlearning methods. To facilitate analysis and discussion, we categorize these into four groups based on their unlearning strategy:

**Fine Tuning** (`FT`): Gradient Descent (`GD`), Successive Random Labels (`SRL`), NegGrad (`NG`) (Golatkar et al. (2020)), Advanced NegGrad (`ANG`) (Choi & Na (2023)), UNSIR (`UNSIR`) (Tarun et al. (2023)). These methods train the model further according to specific strategies.

**Selective Weight Modification** (`SWM`): CF-k (`CFk`) (Goel et al. (2022)), EU-k (`EUk`) (Goel et al. (2022)), Saliency Unlearning (`SalUn`) (Fan et al. (2023)). These methods only operate on a subset of the model parameters, typically the last layers or weights identified via saliency masking.

**Distillation** (`DIS`): Bad Teaching (`BT`) (Chundawat et al. (2023)), SCRUB (`SCRUB`) (Kurmanji et al. (2024)). These methods rely on teacher–student setups to alter the target model's behavior.

**Weight Importance** (`WI`): Fisher Forgetting (`FF`) (Golatkar et al. (2020)), Selective Synaptic Dampening (`SSD`) (Foster et al. (2024)). These directly modify model weights, leveraging the Fisher Information Matrix to estimate the importance of parameters with respect to $D_f$.

For reference, the metrics related to the *Original* (`Orig.`) model and the *Gold Model* (`Gold`) are also reported for each dataset. In a few cases, certain Unlearners required minor modifications (e.g., change of loss function) to work consistently across settings.

## 3.3 METRICS

We evaluate unlearning methods along three key dimensions:

**Utility**: the predictive performance of the model after unlearning to evaluate the unintended degradation on non-forgotten samples. We compute the F1 score on both the test set and the forget set.

**Efficacy**: the extent to which the influence of the forget set is removed. We adopt the Unlearning Membership Inference Attack (UMIA) (Hayes et al., 2024), which tests whether the model can distinguish forgotten samples from previously unseen ones. Effective unlearning yields UMIA values close to those of the Gold model, avoiding both *over-* and *under*-unlearning (Shi et al., 2024).

**Efficiency**: the computational cost of unlearning. We measure the relative training speedup compared to full retraining, as well as the peak GPU memory usage during execution.

In the following, we refer to Utility, Efficacy, and Efficiency as Evaluation Dimensions (*ED*). While we refer to single benchmarking scores chosen for the *ED*s (e.g., the UMIA for Efficacy) as *measures*. Compound metrics will be referred to simply as *metrics*.

### 3.3.1 LUMA: A UNIFIED METRIC

Although examining the three *ED*s separately offers a detailed view of an Unlearner's quality, a unified MU metric is essential for comprehensively comparing methods, whether to discard under-performing hyperparameters or to identify the best performer in real-world applications.

Despite its importance, a unified metric remains a critical gap in the MU literature. Koudounas et al. (2025) introduces GUM, a Global Unlearning Metric, but it faces limitations on scalability to multiple measures and resilience to edge cases. GUM reduces each dimension to a single proxy measure (e.g., UMIA for Efficacy, F1 score for Utility, RunTime for Efficiency), which fails to cap-

ture the richness of MU performance when multiple measures per *ED* are available: in practice, MU performance is evaluated by several measures (e.g., F1 scores on forget and test sets, RunTime and GPU memory usage), and considering all these aspects jointly provides a more faithful assessment than relying on single-value proxies. Zhao et al. (2024) proposed Tug of War (ToW), defined as the product of the relative differences between the Gold and retrained models on accuracies over the test, retain, and forget sets. However, ToW excludes Efficiency, leading to a partial evaluation. We provide empirical evidence of these shortcomings in Appendix A.1.

To address these limitations, we introduce the **Laplacian Unlearning Multidimensional Assessment (LUMA)**. Unlike GUM and ToW, LUMA is multidimensional by design, incorporating vectors of measures within each *ED*, and flexible, allowing users to add any measure deemed relevant. LUMA computes the distance of an unlearned model from the retrained Gold model across utility, efficacy, and efficiency, while penalizing large deviations in any single dimension.

All *ED*s are referenced against the Gold Model (Gold), which represents the ideal target of MU methods (Xu et al. (2024)) as the model retrained from scratch only on the retain set. Let Gold and $M'$ be the Gold Model and the model obtained via the application of the MU method to be measured, respectively.

In our benchmark, the efficacy dimension is measured by UMIA, the utility dimension by F1 score on the test and forget sets, and the Efficiency dimension by runtime and GPU memory usage. Each *ED* can therefore be represented as a vector of the values computed by the chosen measures on the considered model $\bullet \in \{\text{Gold}, M'\}$

$$\mathbf{e}_\bullet = \left[\text{UMIA}_\bullet\right]^\top, \ \mathbf{u}_\bullet = \left[F1_\bullet^{test}, F1_\bullet^{forget}\right]^\top, \ \mathbf{t}_\bullet = \left[\text{RunTime}_\bullet, \text{Memory}_\bullet\right]^\top$$

To compare the Unlearners against the Gold Model, we map efficacy and utility into similarity factors using a $\gamma$-parametrized Laplacian kernel. This choice penalizes *under-* or *over*-performance relative to the Gold Model, while amplifying large deviations in any individual measure.

$$M_U(\mathbf{u}_{\text{Gold}}, \mathbf{u}_{M'}) = \exp((-\gamma\|\mathbf{u}_{\text{Gold}} - \mathbf{u}_{M'}\|_1)),$$
$$M_E(\mathbf{e}_{\text{Gold}}, \mathbf{e}_{M'}) = \exp((-\gamma\|\mathbf{e}_{\text{Gold}} - \mathbf{e}_{M'}\|_1)).$$

While their closeness to the Gold Model evaluates utility and efficacy, efficiency follows a different principle: the less time and memory a method consumes, the better; the Gold Model provides an upper bound reference point. To capture this, we first normalize each efficiency measure by the corresponding value of the Gold Model. We then apply a logarithmic transformation to smooth extreme differences and emphasize relative improvements over absolute ones. Finally, we apply an exponential decay so that larger deviations from the Gold baseline are penalized more strongly, and aggregate the resulting values through a weighted average to obtain a single efficiency score:

$$M_T(\mathbf{t}_{\text{Gold}}, \mathbf{t}_{M'}) = \sum_i \mathbf{w}_i \exp\left[-\left(\frac{\log(1 + \mathbf{t}_{M',i})}{\log(1 + \mathbf{t}_{\text{Gold},i})}\right)^3\right],$$

where $\mathbf{w}_i$ represents the weight assigned to the $i^{th}$ efficiency measure.

Finally, we define the unified metric as:

**Definition 1** (LUMA)**.** *Let $M_U, M_E, M_T$ be the similarity scores for utility, efficacy, and efficiency relative to the Gold model (as defined above). Then*

$$LUMA = \frac{3}{\frac{1}{M_U} + \frac{1}{M_E} + \frac{1}{M_T}}.$$

In other words, LUMA is the harmonic mean of $M_U$, $M_E$, and $M_T$. LUMA is parametrized by the $\gamma$ of the Laplacian kernels and the weight vector $\mathbf{w}$ assigned to the efficiency measures. In our setting, we set $\gamma = 3$ and $\mathbf{w} = (0.9, 0.1)$, assigning 90% of the weight to RunTime and 10% to memory efficiency. We detail parameter tuning of LUMA in Section A.3.

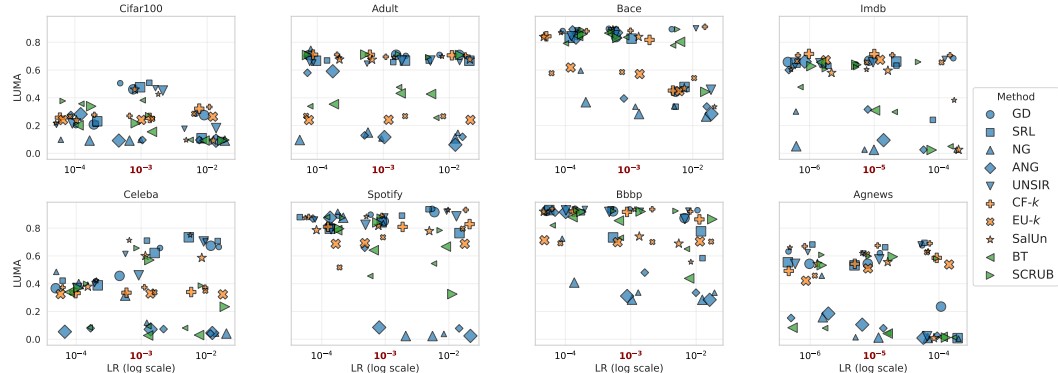

Figure 2: Performance of unlearning methods across datasets under varying learning rates. Each subplot represents a dataset. The marker size denotes the underlying model dimension (small vs. large), while the color indicates the unlearner family (FT, SWM, DIS, WI). The LR used to train the original model is highlighted in dark red.

Note that since the Gold Model is also scored (with a value of 1 for both Utility and Efficacy, as the difference w.r.t. itself is, by definition, 0), its LUMA is fixed. Consequently, any Unlearner that takes longer than the Gold Model will inevitably be penalized by receiving a lower LUMA score, even if it achieves perfect similarity with the Gold Model. This property makes it straightforward to identify cases where an Unlearner is not suitable for application.

LUMA offers several advantages: *(i).* It is designed to range from 0 to 1, where 1 is the ideal MU algorithm that returns a model identical to the Gold Model with no additional cost in time or memory usage. *(ii).* It strongly penalizes any significant deviation on any measure, enabling fast pruning of ill-defined (or ill-parameterized) Unlearners. *(iii).* It is extensible, allowing the integration of any task-specific measure. For example, ROC can replace F1 in the Efficacy *ED*, or be added alongside it without further modifications. This flexibility is unique to LUMA and ensures seamless integration with future evaluation protocols.

## 4 RESULTS

In this Section, we summarize the main findings from our benchmark study. In Section 4.1, we start by studying hyperparameters and we show that the size of the model has little impact on the performance of Unlearners. In Section 4.2, we report the main results from our experimentation on the four domains (Tabular, Image, Textual, Graphs). Finally, Section 4.3 summarizes takeaways.

### 4.1 HYPERPARAMETER TUNING AND MODEL SELECTION

Most of the Unlearners are only parametrized by their Learning Rate (LR) of either the fine-tuning to be applied, the distillation, or any semi-Newton step they employ (refer to Section 3.2 for a taxonomy). In our benchmark, we tuned these parameters by employing LRs one order of magnitude lower, one order of magnitude higher, and equal to the training LR for each domain.

Figure 2 shows the LUMA of each unlearner across datasets and models as the learning rate varies. We use marker size to represent the underlying model dimension. Figure 2 shows that the Negative Gradient unlearners (NG and ANG) are the most sensitive to changes in the LR parameter, often severely degrading performance unless the unlearning LR is set lower than the training LR. In general, using a larger LR than that of training leads to poor LUMA, while the best performance is usually achieved with a rate close to the training value—particularly evident in the Image (CIFAR-100, CelebA) and Text (IMDB, AG News) domains. This makes the tuning of Unlearner dependent on LR quite simple, as the most reliable choice is always to select an LR comparable to the one used for training.

Table 2: Comparison of Unlearners on the Image datasets trained on ResNet-50.

| Group | Method | Cifar 100 | | | | Celeba | | | |
|---|---|---|---|---|---|---|---|---|---|
| | | F1 test | UMIA | Runtime | LUMA | F1 test | UMIA | Runtime | LUMA |
| − | Orig. | .466 ± .007 | .706 ± .028 | 3087.0 ± 442.3 | .266 ± .046 | .688 ± .009 | .770 ± .011 | 8967.0 ± 744.0 | .303 ± .027 |
| − | Gold | .429 ± .003 | .501 ± .003 | 3087.0 ± 442.3 | .636 ± .000 | .670 ± .009 | .539 ± .008 | 8967.0 ± 744.0 | .636 ± .000 |
| FT | GD | .437 ± .001 | .656 ± .034 | 112.3 ± 22.2 | .461 ± .077 | .578 ± .039 | .535 ± .006 | 291.3 ± 31.8 | .673 ± .117 |
| | SRL | .438 ± .009 | .649 ± .035 | 141.5 ± 21.8 | .476 ± .072 | .607 ± .011 | .529 ± .005 | 291.8 ± 32.7 | .734 ± .011 |
| | NG | .001 ± .000 | .499 ± .001 | 38.3 ± 6.1 | .092 ± .004 | .687 ± .004 | .733 ± .006 | 1.5 ± .1 | .397 ± .023 |
| | ANG | .154 ± .011 | .536 ± .016 | 224.1 ± 32.7 | .282 ± .027 | .216 ± .055 | .750 ± .099 | 693.4 ± 26.1 | .071 ± .019 |
| | UNSIR | .436 ± .005 | .652 ± .033 | 317.6 ± 29.6 | .452 ± .071 | .589 ± .044 | .529 ± .008 | 294.1 ± 19.2 | .704 ± .091 |
| SWM | CF$k$ | .462 ± .009 | .693 ± .024 | 89.9 ± 18.7 | .322 ± .029 | .697 ± .001 | .772 ± .008 | 250.1 ± 24.6 | .340 ± .024 |
| | EU$k$ | .476 ± .004 | .697 ± .027 | 1733.0 ± 313.3 | .264 ± .031 | .697 ± .002 | .768 ± .006 | 1269.0 ± 78.4 | .329 ± .013 |
| | SalUn | .434 ± .021 | .659 ± .048 | 153.2 ± 1.6 | .460 ± .147 | .682 ± .005 | .620 ± .020 | 430.3 ± 199.6 | .600 ± .059 |
| DIS | BT | .074 ± .014 | .651 ± .013 | 286.7 ± 5.2 | .202 ± .042 | .318 ± .091 | .618 ± .021 | 744.4 ± 69.6 | .340 ± .128 |
| | SCRUB | .460 ± .025 | .679 ± .038 | 143.2 ± 15.6 | .339 ± .100 | .648 ± .016 | .638 ± .008 | 418.9 ± 126.4 | .570 ± .019 |
| WI | FF | .186 ± .161 | .065 ± .922 | 2340.0 ± 121.8 | .341 ± .294 | .617 ± .033 | .655 ± .024 | 205.6 ± 26.5 | .575 ± .071 |
| | SSD | .466 ± .007 | .681 ± .030 | 120.9 ± .8 | .316 ± .066 | .688 ± .009 | .773 ± .013 | 280.2 ± 40.3 | .350 ± .040 |

Table 3: Comparison of Unlearners on the Tabular datasets trained on MLP.

| Group | Method | Adult | | | | Spotify | | | |
|---|---|---|---|---|---|---|---|---|---|
| | | F1 test | UMIA | Runtime | LUMA | F1 test | UMIA | Runtime | LUMA |
| − | Orig. | .793 ± .002 | .498 ± .001 | 135.7 ± 7.9 | .631 ± .000 | .635 ± .009 | .528 ± .005 | 119.6 ± 1.7 | .573 ± .007 |
| − | Gold | .791 ± .002 | .499 ± .001 | 135.7 ± 7.9 | .636 ± .000 | .629 ± .009 | .497 ± .003 | 119.6 ± 1.7 | .636 ± .000 |
| FT | GD | .791 ± .003 | .498 ± .002 | 88.3 ± 4.9 | .710 ± .001 | .604 ± .007 | .498 ± .004 | 4.7 ± .1 | .915 ± .018 |
| | SRL | .790 ± .002 | .500 ± .000 | 11.2 ± 4.7 | .668 ± .002 | .634 ± .007 | .521 ± .002 | 7.8 ± .1 | .842 ± .012 |
| | NG | .434 ± .000 | .500 ± .002 | 16.2 ± .4 | .150 ± .001 | .604 ± .016 | .522 ± .008 | 1.5 ± .0 | .875 ± .006 |
| | ANG | .767 ± .004 | .499 ± .000 | 140.7 ± 1.5 | .591 ± .006 | .608 ± .009 | .503 ± .003 | 11.9 ± .1 | .875 ± .002 |
| | UNSIR | .792 ± .003 | .498 ± .001 | 105.2 ± .6 | .668 ± .008 | .603 ± .007 | .499 ± .001 | 8.5 ± .2 | .885 ± .012 |
| SWM | CF$k$ | .791 ± .003 | .498 ± .002 | 85.5 ± 1.1 | .711 ± .007 | .642 ± .007 | .519 ± .007 | 5.3 ± .1 | .829 ± .011 |
| | EU$k$ | .793 ± .002 | .498 ± .001 | 848.4 ± 15.6 | .242 ± .010 | .642 ± .007 | .514 ± .007 | 59.6 ± .3 | .692 ± .000 |
| | SalUn | .786 ± .003 | .500 ± .001 | 100.8 ± 11.9 | .676 ± .014 | .635 ± .009 | .526 ± .006 | 8.3 ± .6 | .817 ± .021 |
| DIS | BT | .660 ± .106 | .504 ± .005 | 156.1 ± .9 | .432 ± .154 | .583 ± .027 | .531 ± .003 | 13.2 ± 1.0 | .785 ± .029 |
| | SCRUB | .789 ± .003 | .498 ± .002 | 81.4 ± 1.0 | .711 ± .007 | .611 ± .013 | .515 ± .007 | 7.6 ± .1 | .841 ± .004 |
| WI | FF | .786 ± .007 | .500 ± .001 | 8.0 ± .8 | .936 ± .013 | .600 ± .028 | .517 ± .008 | 19.4 ± .2 | .812 ± .017 |
| | SSD | .793 ± .002 | .499 ± .001 | 91.5 ± .9 | .697 ± .016 | .635 ± .009 | .518 ± .007 | 8.1 ± .1 | .827 ± .017 |

The only Unlearners that take as input a parameter different from the LR are the ones in the group of Weight Importance (WI): Fisher Forgetting (FF) and Selective Synaptic Dampening (SSD). In this case, the tuning was applied to their respective parameters, and its results are reported in Section C of the Appendix. In the remainder of the paper, we only report the best results for each Unlearner.

The second key takeaway from Figure 2 is that the size of the model has little effect on the performance of the Unlearners, as they yield close scores across all *ED*s and, as a result, they are close in terms of LUMA. Unlearners perform similarly on models trained on a specific dataset, regardless of their sizes. This result is consistent across all domains.

As a result, in the remainder of the paper, we will only report results from the bigger of the two models. For completeness, we report all other results in Section C of the Appendix.

## 4.2 MAIN RESULTS

In Tables 2, 3, 4, and 5 we report the main results of our benchmark. All experiments were conducted three times on three different seeds, and the tables report the mean results along with their standard deviation. While LUMA is computed using the full set of measures described in Section 3, for the sake of space we only report one representative measure per *ED* (i.e., test F1 score, UMIA, and RunTime), together with the aggregated LUMA for all experiments. However, full results, including runs for smaller models and different parameters, are reported in Section C. From these tables, we draw intra- (Section 4.2.1) and inter-domain (Section 4.2.2) observations.

### 4.2.1 INTRA-DOMAIN OBSERVATIONS

**Image domain**. Table 2 shows that the best-performing Unlearners in the image domain, according to LUMA, are SRL, GD, SalUn, and UNSIR, which achieve comparable results. This holds across

Table 4: Comparison of the Graph datasets. Each entry reports mean ± std.

| Group | Method | BACE | | | | BBBP | | | |
|---|---|---|---|---|---|---|---|---|---|
| | | F1 test | UMIA | Runtime | LUMA | F1 test | UMIA | Runtime | LUMA |
| – | Orig. | $0.630 \pm 0.004$ | $0.525 \pm 0.025$ | $57.3 \pm 27.8$ | $0.564 \pm 0.048$ | $0.702 \pm 0.003$ | $0.501 \pm 0.007$ | $55.1 \pm 0.2$ | $0.617 \pm 0.002$ |
| – | Gold | $0.559 \pm 0.044$ | $0.528 \pm 0.009$ | $57.3 \pm 27.8$ | $0.636 \pm 0.000$ | $0.712 \pm 0.007$ | $0.498 \pm 0.005$ | $55.1 \pm 0.2$ | $0.636 \pm 0.000$ |
| FT | GD | $0.612 \pm 0.021$ | $0.519 \pm 0.017$ | $3.1 \pm 4.0$ | $0.851 \pm 0.102$ | $0.705 \pm 0.005$ | $0.499 \pm 0.006$ | $1.0 \pm 0.0$ | $0.924 \pm 0.014$ |
| | SRL | $0.609 \pm 0.015$ | $0.536 \pm 0.017$ | $4.0 \pm 5.1$ | $0.832 \pm 0.089$ | $0.710 \pm 0.005$ | $0.496 \pm 0.004$ | $1.4 \pm 0.0$ | $0.930 \pm 0.013$ |
| | NG | $0.385 \pm 0.026$ | $0.488 \pm 0.011$ | $0.2 \pm 0.0$ | $0.368 \pm 0.033$ | $0.485 \pm 0.046$ | $0.495 \pm 0.001$ | $0.3 \pm 0.0$ | $0.408 \pm 0.126$ |
| | ANG | $0.534 \pm 0.028$ | $0.496 \pm 0.017$ | $0.7 \pm 0.0$ | $0.869 \pm 0.031$ | $0.688 \pm 0.012$ | $0.498 \pm 0.011$ | $0.9 \pm 0.0$ | $0.916 \pm 0.022$ |
| | UNSIR | $0.617 \pm 0.009$ | $0.528 \pm 0.006$ | $1.0 \pm 0.0$ | $0.850 \pm 0.097$ | $0.705 \pm 0.005$ | $0.494 \pm 0.006$ | $1.3 \pm 0.0$ | $0.928 \pm 0.009$ |
| SWM | CF$k$ | $0.612 \pm 0.021$ | $0.522 \pm 0.017$ | $3.2 \pm 4.1$ | $0.843 \pm 0.112$ | $0.705 \pm 0.005$ | $0.488 \pm 0.002$ | $1.1 \pm 0.0$ | $0.920 \pm 0.008$ |
| | EU$k$ | $0.616 \pm 0.021$ | $0.537 \pm 0.015$ | $42.9 \pm 0.2$ | $0.618 \pm 0.110$ | $0.709 \pm 0.001$ | $0.492 \pm 0.006$ | $34.0 \pm 0.0$ | $0.713 \pm 0.005$ |
| | SalUn | $0.609 \pm 0.015$ | $0.536 \pm 0.023$ | $1.3 \pm 0.0$ | $0.839 \pm 0.101$ | $0.711 \pm 0.007$ | $0.497 \pm 0.006$ | $1.7 \pm 0.1$ | $0.919 \pm 0.012$ |
| DIS | BT | $0.570 \pm 0.045$ | $0.529 \pm 0.041$ | $2.2 \pm 0.1$ | $0.866 \pm 0.011$ | $0.659 \pm 0.010$ | $0.494 \pm 0.008$ | $2.9 \pm 0.0$ | $0.879 \pm 0.028$ |
| | SCRUB | $0.612 \pm 0.021$ | $0.523 \pm 0.013$ | $1.4 \pm 0.0$ | $0.857 \pm 0.126$ | $0.717 \pm 0.008$ | $0.498 \pm 0.004$ | $1.9 \pm 0.0$ | $0.920 \pm 0.032$ |
| WI | FF | $0.607 \pm 0.020$ | $0.526 \pm 0.017$ | $6.9 \pm 0.1$ | $0.830 \pm 0.108$ | $0.669 \pm 0.066$ | $0.505 \pm 0.008$ | $7.0 \pm 0.0$ | $0.793 \pm 0.147$ |
| | SSD | $0.630 \pm 0.004$ | $0.527 \pm 0.009$ | $1.2 \pm 0.0$ | $0.823 \pm 0.099$ | $0.702 \pm 0.003$ | $0.497 \pm 0.003$ | $1.6 \pm 0.1$ | $0.935 \pm 0.006$ |

Table 5: Comparison of Unlearners on the Textual datasets trained on BERT.

| Group | Method | IMDB | | | | AG news | | | |
|---|---|---|---|---|---|---|---|---|---|
| | | F1 test | UMIA | Runtime | LUMA | F1 test | UMIA | Runtime | LUMA |
| – | Orig. | $.937 \pm .005$ | $.549 \pm .004$ | $4914.0 \pm 2965.0$ | $.565 \pm .006$ | $.880 \pm .012$ | $.641 \pm .030$ | $5200.0 \pm 54.7$ | $.399 \pm .056$ |
| – | Gold | $.939 \pm .006$ | $.501 \pm .000$ | $4914.0 \pm 2965.0$ | $.636 \pm .000$ | $.906 \pm .005$ | $.525 \pm .024$ | $5200.0 \pm 54.7$ | $.636 \pm .000$ |
| FT | GD | $.941 \pm .002$ | $.539 \pm .001$ | $1329.0 \pm 4.1$ | $.667 \pm .045$ | $.908 \pm .003$ | $.568 \pm .009$ | $1655.0 \pm 24.9$ | $.544 \pm .063$ |
| | SRL | $.939 \pm .001$ | $.542 \pm .003$ | $1402.0 \pm .2$ | $.661 \pm .043$ | $.913 \pm .001$ | $.545 \pm .006$ | $1748.0 \pm .4$ | $.554 \pm .061$ |
| | NG | $.408 \pm .090$ | $.497 \pm .000$ | $75.9 \pm 4.8$ | $.051 \pm .031$ | $.570 \pm .004$ | $.852 \pm .000$ | $65.3 \pm .1$ | $.158 \pm .028$ |
| | ANG | $.935 \pm .003$ | $.502 \pm .010$ | $271.0 \pm 8.4$ | $.660 \pm .056$ | $.893 \pm .002$ | $.975 \pm .001$ | $3414.0 \pm 8.4$ | $.185 \pm .006$ |
| | UNSIR | $.933 \pm .010$ | $.538 \pm .004$ | $1478.0 \pm 6.1$ | $.656 \pm .036$ | $.910 \pm .002$ | $.554 \pm .015$ | $1814.0 \pm .7$ | $.575 \pm .095$ |
| SWM | CF$k$ | $.938 \pm .005$ | $.549 \pm .006$ | $512.7 \pm 2.9$ | $.716 \pm .038$ | $.905 \pm .005$ | $.555 \pm .017$ | $599.1 \pm 5.9$ | $.586 \pm .076$ |
| | EU$k$ | $.939 \pm .004$ | $.551 \pm .002$ | $1025.0 \pm 2.5$ | $.674 \pm .042$ | $.907 \pm .004$ | $.556 \pm .014$ | $1785.0 \pm 17.2$ | $.538 \pm .059$ |
| | SalUn | $.926 \pm .000$ | $.540 \pm .000$ | $1988.0 \pm 0.0$ | $.596 \pm .047$ | $.912 \pm .001$ | $.552 \pm .006$ | $2154.0 \pm 1.8$ | $.556 \pm .047$ |
| DIS | BT | $.890 \pm .000$ | $.540 \pm .000$ | $2427.0 \pm 0.0$ | $.656 \pm .000$ | $.289 \pm .044$ | $.784 \pm .140$ | $299.0 \pm 78.3$ | $.083 \pm .027$ |
| | SCRUB | $.935 \pm .009$ | $.543 \pm .001$ | $1904.0 \pm 132.5$ | $.638 \pm .055$ | $.888 \pm .011$ | $.563 \pm .003$ | $2258.0 \pm 13.4$ | $.594 \pm .087$ |
| WI | FF | $.500 \pm .011$ | $.498 \pm .002$ | $89.2 \pm 3.3$ | $.086 \pm .000$ | $.142 \pm .058$ | $.854 \pm .007$ | $.3 \pm .0$ | $.009 \pm .001$ |
| | SSD | $.941 \pm .000$ | $.555 \pm .015$ | $1492.0 \pm 83.8$ | $.642 \pm .065$ | $.879 \pm .010$ | $.654 \pm .012$ | $1793.0 \pm 46.3$ | $.440 \pm .065$ |

both datasets, despite their different setups (CIFAR-100 for multiclass and CelebA for multilabel classification), indicating a degree of stability in image-domain Unlearners. This is not surprising, as the image domain is by far the most extensively studied (see Section 2). However, all these methods substantially increase UMIA relative to Gold (up to .659 and .623, compared to .501 and .539), which results in relatively low LUMA values (below 0.5 and lower than Gold). This highlights that the problem remains unsolved: current Unlearners have yet to match gold-level performance without compromising UMIA.

**Tabular domain.** Table 3 shows that FF is the clear winner on the Adult dataset, followed by GD and SSD. These methods also achieve strong performance on the Spotify dataset. In general, WI methods perform very well in this domain, due to the simpler architecture of MLPs. Both FF and SSD compute a Fisher Information Matrix over the network, which is reasonably accurate on an MLP (Karakida & Osawa (2020)). This is corroborated by the fact that Unlearners in the tabular domain obtain substantially higher LUMA values, reflecting the relative simplicity of this setting. Fine-tuning (FT) methods are also effective, except for NG, which consistently stays behind.

**Graph domain.** FT and DIS Unlearners perform best in the Graph domain, as shown in Table 4. All methods exhibit extremely low running times and relatively minor deviations from the F1 test, as GCNs are notoriously able to generalize better even with fewer samples (Yang et al. (2023)), so they aren't impacted as much by the removal of the Forget Set, and the LUMA scores are very high across the board. This holds across both datasets, although it also highlights the difficulty of completely erasing information from the Gold model itself. Overall, unlearning for graph classification remains underexplored and warrants further investigation.

**Textual domain.** Textual domain is where SWM methods shine, as reported in Table 5. This is because BERT (or LLMs in general) usually fine-tune the last classification layer, while the bulk of knowledge is encoded within the deeper transformer architecture. As such, methods that selectively modify weights (CF$k$ EU$k$, SalUn) remove task-specific information without harming the general representations captured by the transformer layers. Interestingly, WI Unlearners are unable to lever-

age the Fisher Information Matrix to correctly identify which weights to modify on LLMs, as shown by the low LUMA score for `FF` and `SSD` on AG news. Lastly, `FT` and `DIS` methods achieve good performance (with a few exceptions) but are held back by the non-trivial cost of further training a model of this size in its entirety.

### 4.2.2 INTER-DOMAIN OBSERVATIONS

No single Unlearner (or group of Unlearners) dominates across all modalities, which reflects both the early stage of MU research and the ongoing search for more reliable methods. Still, some consistent patterns can be observed across domains.

Fine-tuning is often a safe strategy, so `FT` Unlearners are generally the most reliable, with the notable exception of `NG`, which often performs worst in terms of LUMA. Because the Forget Sets we define are heterogeneous (spanning multiple classes), a single epoch of negative gradient updates tends to collapse the model, leading to poor F1 scores on the Test Set. `ANG`, by contrast, mitigates this issue by also taking the Retain Set into account, and should therefore be always preferred.

Similarly, `SWM` Unlearners perform consistently well, but they rarely achieve the best performance. Among these, `EUk` is consistently the worst in terms of LUMA, while `SalUn` is the most reliable across all domains. This aligns with its widespread adoption in the literature, where it often serves as a reference baseline.

On the other hand, both `DIS` and `WI` Unlearners show inconsistent performance. The former group performs best with smaller models (e.g., the tabular and graph domains). Still, it fails to scale to large pretrained architectures such as ResNet-50 and BERT, as distilling knowledge from such complex models is considerably more challenging, which is consistent with the literature (Marrie et al. (2024); Fang et al. (2025)). The latter, which directly modify model parameters, are particularly effective given the simpler architecture of MLPs compared to CNNs (e.g., ResNet50) or LLMs (e.g., BERT), where kernels and attention mechanisms may introduce unintended side effects. Among these, `SSD` is generally more reliable.

### 4.3 MAIN TAKEAWAYS

The main results of our benchmark can be summarized as follows: ***(i).*** Unlearners parameterized by learning rate for a semi-Newton step should adopt a value close to that used during training. Setting the learning rate too high leads to severe degradation of utility, effectively rendering the model unusable. ***(ii).*** Scaling up model size does not help: large models do not mitigate unlearning weaknesses. ***(iii).*** Unlearning remains fundamentally domain-dependent, as no method succeeds across all domains, although Fine-tuning Unlearners are the most reliable. ***(iv).*** A utility–efficacy trade-off is unavoidable with current methods (especially so in the Image domain), showing the field lacks methods that can forget without leaking information. ***(v).*** Distillation Unlearning is not scalable: it fails on large pretrained models, while selective weight methods are more reliable for these models. ***(vi).*** LUMA exposes hidden weaknesses: prior benchmarks overstated progress by ignoring efficiency and efficacy simultaneously.

## 5 CONCLUSION

In this work, we bring order to the landscape of Machine Unlearning methods for classification by providing: *(i).* the most comprehensive benchmark to date, covering 4 data modalities, 12 Unlearners, 8 datasets, and 8 models; *(ii).* the first systematic evaluation on the tabular and graph modalities; *(iii).* a taxonomy of Unlearner methods in the literature, *(iv).* a novel unified metric, LUMA, which quantifies performance as a single value to support hyperparameter tuning and unlearner selection in practice; and *(v).* a publicly available, fully reproducible, and extensible benchmark to facilitate fair comparison in future work.

Our results lead to clear guidelines for the design and deployment of Unlearners, and expose critical shortcomings that can only be revealed through cross-modality analysis. Together, these contributions establish a common ground for evaluating and comparing Machine Unlearning approaches for classification. By ensuring reproducibility and extensibility, our work provides a solid basis for future methods to be rigorously evaluated.

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

# LLM USAGE DISCLOSURE

During the preparation of this work, the authors used LLMs to correct typos and grammatical mistakes. After using this tool/service, the authors reviewed and edited the content as needed and take full responsibility for the content of the published article.

# A   LUMA AGAINST OTHER METRICS

In this section, we show shortcomings and issues of the previous two unified unlearning metrics, GUM(Koudounas et al. (2025)) and ToW (Zhao et al. (2024)). Then, we detail how LUMA handles edge cases and its sensitivity to hyperparameters.

## A.1   SHORTCOMINGS OF OTHER METRICS

Indicating the Gold Model with $g$ and the resulting Unlearned model with $u$, GUM is defined as:

$$\text{GUM} = \frac{(1 + \alpha + \beta)UET}{\alpha ET + \beta UT + UE}$$

where $U = 1 - \left| F1_T^{(g)} - F1_T^{(u)} \right|, \quad E = 1 - \left( \frac{\text{MIA}'(u) - \text{MIA}'(g)}{\text{MIA}(o) - \text{MIA}'(g)} \right)^2$, with

$$\text{MIA}'(u) = \min\{\text{MIA}(u), \text{MIA}(o)\}, \quad \text{MIA}'(g) = \min\left\{ \text{MIA}(g), \frac{\text{MIA}'(u) + \text{MIA}(o)}{2} \right\}.$$

and lastly $T = 1 - \frac{\log\left( T^{(u)} + 1 \right)}{\log\left( T^{(g)} + 1 \right)}$.

While GUM includes all the three key evaluation dimensions (*ED*s) for MU methods (efficacy, utility, efficiency) it suffers from two core issues: *(i)*. Only using one measure per dimension: the F1 score on the test set is not enough to assess the model's utility, and the RunTime is not enough to assess the MU method's efficiency. The F1 score on the forget set and the memory efficiency are also important. *(ii)*. When $MIA'(u) - MIA(o) \leq \epsilon$ with a reasonably small $\epsilon$, $E$ grows without bound, effectively dominating GUM to unusable values. Worse, the metric is not computable at all for a division by 0 given two conditions:

- $MIA(g) \geq MIA(o)$
- $MIA'(u) \geq MIA(o)$

Consider the definition of $E$, specifically the denominator $MIA(o) - MIA'(g)$. This will be 0 when $MIA(o) = MIA'(g)$. Given the definition of $MIA'(g)$, this can only happen when $MIA(g) > MIA(o)$ (first condition) and $MIA'(u) = MIA(o)$. The latter is true when $MIA(u) \geq MIA(o)$ (second condition). We show an example in Section A.2.

Tug of War (ToW) is instead defined as:

$$\text{ToW}(\theta_u, \theta_g, S, R, D_{\text{test}}) = \left(1 - d_a(\theta_u, \theta_g, S)\right) \left(1 - d_a(\theta_u, \theta_g, R)\right) \left(1 - d_a(\theta_u, \theta_g, D_{\text{test}})\right),$$

where

$$a(\theta, D) = \frac{1}{|D|} \sum_{(x,y) \in D} \mathbf{1}[f(x; \theta) = y]$$

is the accuracy of a model $f$ parameterized by $\theta$ on dataset $D$, and

$$d_a(\theta_u, \theta_r, D) = \left| a(\theta_u, D) - a(\theta_r, D) \right|$$

is the absolute difference between the accuracies of models $\theta_u$ (the unlearned model) and $\theta_g$ (the Gold Model) on the dataset $D$.

ToW includes multiple metrics for the Utility dimension, but misses the Efficiency dimension entirely. Accordingly, the Gold Model will always obtain the optimal ToW score of 1 (regardless of how expensive it is to train) and two unlearners that output the same retrained model with largely varying RunTimes will obtain the same ToW score. Moreover, by omitting MIA, ToW doesn't capture the information leakage of any MU method, which is a known problem in the literature (Xu et al. (2024); Le Quy et al. (2022)).

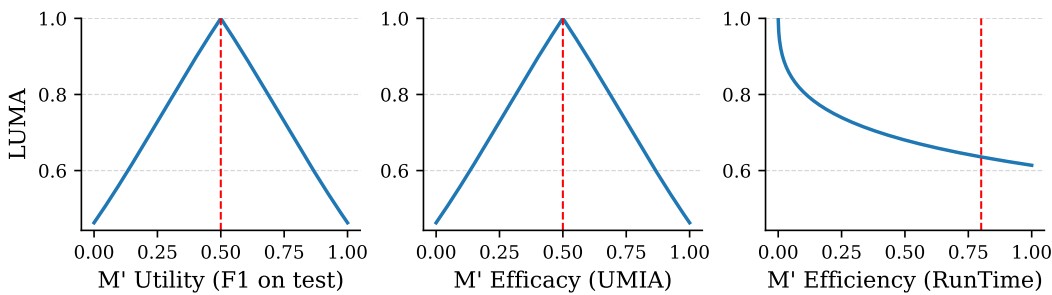

Figure 3: Sensitivity of the proposed LUMA metric with respect to each *ED*.

## A.2 EMPIRICAL EVIDENCE

Table 6: Extract of results on the Adult dataset (Table 3).

| Group | Method | adult3mlp | | | | |
|---|---|---|---|---|---|---|
| | | F1 test | UMIA | Runtime | LUMA | GUM |
| − | Orig. | $.793 \pm .002$ | $.498 \pm .001$ | $135.7 \pm 7.9$ | $.631 \pm .000$ | X |
| − | Gold | $.791 \pm .002$ | $.499 \pm .001$ | $135.7 \pm 7.9$ | $.636 \pm .000$ | X |
| FT | GD | $.791 \pm .003$ | $.498 \pm .002$ | $88.3 \pm 4.9$ | $.710 \pm .001$ | X |

Table 6 is an extract from the bigger Table 3 shown in Section 4. This is empirical evidence of the proven problem of GUM in Section A.1: we have that $MIA(g) \geq MIA(o)$ as $0.499 \geq 0.498$ (first condition) and that $MIA'(u) \geq MIA(o)$ as $0.498 \geq 0.498$. For this reason, GUM cannot be calculated and will fail because of a statistical variation.

GUM is very useful because it combines measures across all *ED*s, but suffer from numerical instability under some conditions. LUMA fixes this by employing Laplacian kernels.

ToW suffers from the opposite problem: ignoring Efficiency entirely. Consider a toy Unlearner that simply retrains the model from scratch, but deliberately taking double the time to do so. Despite the wasted resources, it would still achieve a perfect ToW score. This illustrates that, to serve as a truly unified metric for unlearning, the *ED*dimension of Efficiency must be incorporated.

## A.3 LUMA'S BEHAVIOR AND HYPERPARAMETERS

In contrast, LUMA fixes all shortcomings by considering of all the three dimensions, possibly with more than one metric each. By employing Laplacian kernels, LUMA is easily extendable with future MU measures.

In this Section, we analyze the impact of each *ED* on LUMA, to shed light on its sensitivity and robustness.

By construction, the Laplacian kernels ensure that both positive and negative drifts in an evaluation dimension are penalized symmetrically. Figure 4 illustrates this property, showing the response of LUMA with respect to each *ED* while assuming perfect performance on the remaining *ED*s. For Utility and Efficacy, LUMA reaches its maximum value of 1 when the varying *ED* matches the performance of Gold, and decreases smoothly as the model drifts away in either direction. For Efficiency, LUMA decreases inversely with runtime, lowering as runtime increases.

The Laplacian kernels are parametrized by $\gamma$. In this work, we chose $\gamma = 3$ to severely punish even single-measure drifts between the Gold Model (Gold) and the unlearned Model ($M'$).

Figure 4 illustrates the impact of the kernel parameter $\gamma$ on LUMA as $M'$ drifts from 0 to 1 across all measures. For reference, the Gold model is fixed at 0.5 on all measures. In this work, we chose $\gamma = 3$ as it provided the best smoothness.

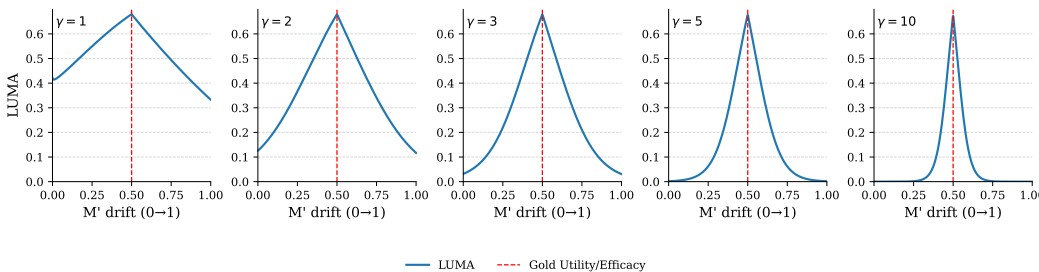

Figure 4: Effect of the Laplacian kernel parameter $\gamma$ on the LUMA metric as a model drifts from an all-zero representation to an all-one representation of the *ED*s of Utility and Efficacy.

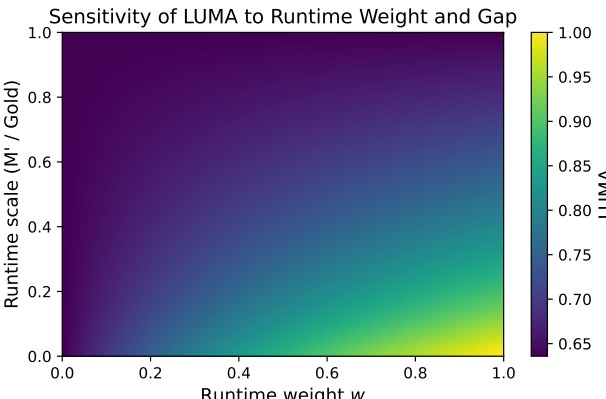

Figure 5: Heatmap of LUMA values as a function of runtime weight $w$ (x-axis) and runtime scale of $M'$ relative to the Gold model (y-axis). The weight assigned to memory is $1 - w$.

Moreover, LUMA is customizable in terms of weights assigned to measures in the Efficiency *ED*. The vast majority of works in the Machine Unlearning literature only consider RunTime when evaluating Efficiency, discarding peak memory usage. In this work, we introduced memory usage as a measure with weight = 0.1, to keep RunTime as the most important efficiency metric. This weight vector can be customized according to available resources and the task at hand. Figure 5 shows the sensitivy of LUMA to the weight vector $w$ when the runtime scales w.r.t. to the Gold Model, assuming equality on the Utility and Efficacy *ED*s. The figure corroborates the correctness of LUMA: if we assign weight 1 to RunTime, and the RunTime is 0, LUMA will be 1, although this is basically unachievable in practice. The vector $w$, weighting Efficiency measures, is customizable. In this study we set $w = [0.9, 0.1]$, as runtime is the dominant Efficiency factor in unlearning. Figure 5 shows that this choice gives runtime decisive influence over the score while still ensuring memory is not ignored.

Summing up, *(i).* By employing Laplacian kernels, LUMA fixes the shortcomings of GUM for edge cases, and surpasses both GUM and ToW in considering more than one measure per *ED* and being easily extensible with other metrics. *(ii.)* By incorporating all *ED*s, LUMA surpasses ToW in completeness, as the latter did not consider Efficiency at all.

## B   BENCHMARK DETAILS

In this section, we show complete details on the Datasets, Forget Sets, Models, and Unlearners we employed in our benchmark. Moreover, in Section B.3, we show how to easily extend the benchmark for future works.

All details and a step-by-step guide can be found at this anonymized repository.

| Dataset | Domain | Features | Samples | Classes |
|---------|--------|----------|---------|---------|
| Adult | Tabular | 14 tabular features | 48,842 | 2 |
| Spotify | Tabular | 15 tabular features | 114,000 | 8 |
| Cifar100 | Image | 32x32 color images | 60,000 | 100 |
| CelebA | Image | 178×218 color images | 202,599 | 40 (ml*) |
| AG news | Text | Textual descriptions | 127,600 | 4 |
| IMDB | Text | Textual descriptions | 100,000 | 2 |
| BACE | Graph | Graphs ($\sim$34N, $\sim$74E, 9F) | 1513 | 2 |
| BBBP | Graph | Graphs ($\sim$23N, $\sim$51E, 9F) | 2,050 | 2 |

Table 7: Overview of the datasets employed for our benchmark. *ml stands for Multilabel. For Graphs, N stands for nodes, E for edges and F for features.

| Architecture | Domain | Epochs | Optimizer | LR |
|--------------|--------|--------|-----------|-----|
| 1-hidden layer MLP | Tabular | 2 | Adam | 1e-3 |
| 3-hidden layer MLP | Tabular | 2 | Adam | 1e-3 |
| ResNet-18 | Image | 5 | Adam | 1e-3 |
| ResNet-50 | Image | 5 | Adam | 1e-3 |
| DistilBERT | Text | 3 | AdamW | 2e-5 |
| BERT | Text | 3 | AdamW | 2e-5 |
| DistilBERT | Text | 3 | AdamW | 2e-5 |
| 1-layer GCN | Graph | 50 | RMSprop | 1e-3 |
| 2-layer GCN | Graph | 50 | RMSprop | 1e-3 |

Table 8: Overview of the models employed for our benchmark.

## B.1 DATASETS

Table 7 shows number of samples, number of attributes, number of classes and domain for each dataset. In the Image domain, we employed CIFAR-100 for multiclass classification and CelebA for binary multilabel classification. As detailed in Section 3, we selected these datasets as the most prominent in the Unlearning literature for each domain.

## B.2 MODELS

Table 8 shows the general architecture of the model we employed. Our rationale was choosing two models per domain with a similar architecture but different in size. Full configurations can be found at this anonymized repository.

## B.3 EXTENDING THE FRAMEWORK

Our benchmark was built to be easily extendable by researchers and practitioners. New experiments can be defined by just extending the configuration files. A full, step-by-step demonstration on how to implement a new Unlearner method is available at this anonymized repository.

## C   COMPLETE RESULTS

In this Section, we report the tables with the complete results across all experiments.

Table 9: Comparison of Unlearners on ResNet18 trained on Cifar100 (mean ± std).

| Group | Method | LR | LUMA | UMIA | F1 (test) | F1 (forget) | RunTime | GPU (MB) |
|-------|--------|-----|------|------|-----------|-------------|---------|----------|
| − | Orig. | - | $.307 \pm .016$ | $.651 \pm .019$ | $.489 \pm .009$ | $.750 \pm .036$ | $1017.0 \pm 289.0$ | $7233 \pm 0$ |
| − | Gold | - | $.636 \pm .000$ | $.499 \pm .001$ | $.420 \pm .016$ | $.421 \pm .014$ | $1017.0 \pm 289.0$ | $7233 \pm 0$ |
| FT | GD | 1e-04 | $.203 \pm .007$ | $.684 \pm .014$ | $.548 \pm .009$ | $.890 \pm .023$ | $75.5 \pm 6.4$ | $8768 \pm 0$ |
| | GD | 1e-03 | $.505 \pm .005$ | $.619 \pm .017$ | $.451 \pm .017$ | $.642 \pm .016$ | $78.4 \pm 9.6$ | $8271 \pm 0$ |
| | GD | 1e-02 | $.286 \pm .067$ | $.502 \pm .002$ | $.155 \pm .056$ | $.156 \pm .052$ | $75.3 \pm 8.3$ | $7772 \pm 0$ |
| | SRL | 1e-04 | $.213 \pm .007$ | $.679 \pm .016$ | $.542 \pm .012$ | $.877 \pm .023$ | $110.1 \pm 7.5$ | $10256 \pm 0$ |
| | SRL | 1e-03 | $.508 \pm .039$ | $.610 \pm .021$ | $.458 \pm .007$ | $.634 \pm .037$ | $109.0 \pm 7.2$ | $9759 \pm 0$ |
| | SRL | 1e-02 | $.096 \pm .011$ | $.499 \pm .000$ | $.000 \pm .000$ | $.000 \pm .000$ | $112.9 \pm 9.2$ | $9267 \pm 0$ |
| | NG | 1e-04 | $.098 \pm .011$ | $.500 \pm .001$ | $.001 \pm .000$ | $.001 \pm .001$ | $23.3 \pm 2.6$ | $13348 \pm 0$ |
| | NG | 1e-03 | $.097 \pm .011$ | $.500 \pm .003$ | $.000 \pm .000$ | $.000 \pm .000$ | $20.8 \pm 1.0$ | $12856 \pm 0$ |
| | NG | 1e-02 | $.097 \pm .011$ | $.500 \pm .000$ | $.000 \pm .000$ | $.000 \pm .000$ | $21.9 \pm 1.1$ | $12366 \pm 0$ |
| | ANG | 1e-04 | $.276 \pm .030$ | $.539 \pm .024$ | $.151 \pm .018$ | $.166 \pm .034$ | $150.6 \pm 24.5$ | $16188 \pm 0$ |
| | ANG | 1e-03 | $.096 \pm .011$ | $.503 \pm .003$ | $.001 \pm .001$ | $.001 \pm .000$ | $156.6 \pm 1.5$ | $15701 \pm 0$ |
| | ANG | 1e-02 | $.096 \pm .011$ | $.501 \pm .002$ | $.000 \pm .000$ | $.000 \pm .000$ | $149.2 \pm 7.2$ | $15206 \pm 0$ |
| | UNSIR | 1e-04 | $.205 \pm .008$ | $.677 \pm .012$ | $.549 \pm .014$ | $.881 \pm .017$ | $183.9 \pm 8.7$ | $19149 \pm 0$ |
| | UNSIR | 1e-03 | $.469 \pm .029$ | $.620 \pm .023$ | $.459 \pm .004$ | $.649 \pm .036$ | $190.1 \pm 4.5$ | $18659 \pm 0$ |
| | UNSIR | 1e-02 | $.177 \pm .054$ | $.502 \pm .002$ | $.083 \pm .036$ | $.084 \pm .036$ | $184.1 \pm 7.6$ | $18172 \pm 0$ |
| SWM | CF$-k$ | 1e-04 | $.277 \pm .016$ | $.658 \pm .017$ | $.520 \pm .006$ | $.816 \pm .031$ | $76.6 \pm 8.5$ | $10498 \pm 0$ |
| | CF$-k$ | 1e-03 | $.270 \pm .012$ | $.658 \pm .016$ | $.524 \pm .007$ | $.821 \pm .029$ | $82.4 \pm 11.9$ | $10228 \pm 0$ |
| | CF$-k$ | 1e-02 | $.334 \pm .024$ | $.645 \pm .016$ | $.498 \pm .004$ | $.771 \pm .033$ | $76.4 \pm 6.7$ | $10225 \pm 0$ |
| | EU$-k$ | 1e-04 | $.266 \pm .011$ | $.647 \pm .015$ | $.516 \pm .006$ | $.799 \pm .025$ | $752.4 \pm 26.0$ | $11172 \pm 0$ |
| | EU$-k$ | 1e-03 | $.248 \pm .008$ | $.653 \pm .015$ | $.525 \pm .008$ | $.815 \pm .026$ | $796.7 \pm 11.4$ | $10903 \pm 0$ |
| | EU$-k$ | 1e-02 | $.285 \pm .017$ | $.647 \pm .016$ | $.503 \pm .007$ | $.781 \pm .032$ | $794.5 \pm 67.7$ | $10768 \pm 0$ |
| | SalUn | 1e-04 | $.218 \pm .008$ | $.677 \pm .016$ | $.539 \pm .013$ | $.872 \pm .021$ | $128.2 \pm 8.6$ | $23841 \pm 0$ |
| | SalUn | 1e-03 | $.426 \pm .062$ | $.629 \pm .028$ | $.474 \pm .012$ | $.687 \pm .060$ | $145.7 \pm 14.1$ | $23257 \pm 0$ |
| | SalUn | 1e-02 | $.096 \pm .011$ | $.500 \pm .000$ | $.000 \pm .000$ | $.000 \pm .000$ | $136.0 \pm 4.6$ | $22671 \pm 0$ |
| DIS | BT | 1e-04 | $.357 \pm .099$ | $.570 \pm .005$ | $.168 \pm .039$ | $.248 \pm .052$ | $190.2 \pm 14.7$ | $18055 \pm 0$ |
| | BT | 1e-03 | $.382 \pm .110$ | $.569 \pm .008$ | $.177 \pm .033$ | $.268 \pm .054$ | $186.5 \pm 10.0$ | $17474 \pm 0$ |
| | BT | 1e-02 | $.097 \pm .012$ | $.500 \pm .001$ | $.001 \pm .001$ | $.002 \pm .002$ | $187.3 \pm 11.1$ | $16896 \pm 0$ |
| | SCRUB | 1e-04 | $.377 \pm .018$ | $.637 \pm .012$ | $.478 \pm .017$ | $.739 \pm .030$ | $104.1 \pm 5.0$ | $19902 \pm 0$ |
| | SCRUB | 1e-03 | $.274 \pm .174$ | $.525 \pm .012$ | $.131 \pm .105$ | $.155 \pm .128$ | $102.6 \pm 12.4$ | $19408 \pm 0$ |
| | SCRUB | 1e-02 | $.096 \pm .011$ | $.499 \pm .000$ | $.000 \pm .000$ | $.000 \pm .000$ | $98.9 \pm 6.9$ | $18916 \pm 0$ |
| WI | FF | 1e-08 | $.322 \pm .195$ | $.548 \pm .044$ | $.168 \pm .147$ | $.229 \pm .201$ | $912.0 \pm 5.4$ | $1877 \pm 0$ |
| | FF | 1e-07 | $.360 \pm .221$ | $.550 \pm .044$ | $.192 \pm .166$ | $.258 \pm .224$ | $935.9 \pm 44.5$ | $1428 \pm 0$ |
| | FF | 1e-06 | $.359 \pm .221$ | $.549 \pm .044$ | $.189 \pm .164$ | $.257 \pm .223$ | $999.0 \pm 148.4$ | $909 \pm 0$ |
| | SSD | 1e-04 | $.357 \pm .022$ | $.638 \pm .016$ | $.489 \pm .009$ | $.750 \pm .036$ | $116.8 \pm 2.5$ | $22130 \pm 0$ |
| | SSD | 1e-03 | $.356 \pm .023$ | $.639 \pm .016$ | $.489 \pm .009$ | $.750 \pm .036$ | $119.4 \pm 6.6$ | $21731 \pm 0$ |
| | SSD | 1e-02 | $.356 \pm .023$ | $.639 \pm .016$ | $.489 \pm .009$ | $.750 \pm .036$ | $119.8 \pm 7.2$ | $21331 \pm 0$ |

Table 10: Comparison of Unlearners on ResNet50 trained on Cifar100 (mean $\pm$ std).

| Group | Method | LR | LUMA | UMIA | F1 (test) | F1 (forget) | RunTime | GPU (MB) |
|---|---|---|---|---|---|---|---|---|
| − | Orig. | - | $.266_{\pm.046}$ | $.706_{\pm.028}$ | $.466_{\pm.007}$ | $.830_{\pm.071}$ | $3087.0_{\pm442.3}$ | $17695_{\pm0}$ |
| − | Gold | - | $.636_{\pm.000}$ | $.501_{\pm.003}$ | $.429_{\pm.003}$ | $.425_{\pm.008}$ | $3087.0_{\pm442.3}$ | $17695_{\pm0}$ |
| FT | GD | 1e-04 | $.208_{\pm.021}$ | $.729_{\pm.032}$ | $.509_{\pm.008}$ | $.921_{\pm.041}$ | $108.9_{\pm21.3}$ | $20924_{\pm0}$ |
| | GD | 1e-03 | $.461_{\pm.077}$ | $.656_{\pm.034}$ | $.437_{\pm.001}$ | $.698_{\pm.067}$ | $112.3_{\pm22.2}$ | $19880_{\pm0}$ |
| | GD | 1e-02 | $.275_{\pm.101}$ | $.500_{\pm.004}$ | $.153_{\pm.062}$ | $.153_{\pm.060}$ | $109.4_{\pm19.8}$ | $18836_{\pm0}$ |
| | SRL | 1e-04 | $.231_{\pm.015}$ | $.726_{\pm.030}$ | $.494_{\pm.013}$ | $.897_{\pm.035}$ | $149.3_{\pm29.4}$ | $24065_{\pm1}$ |
| | SRL | 1e-03 | $.476_{\pm.072}$ | $.649_{\pm.035}$ | $.438_{\pm.009}$ | $.680_{\pm.062}$ | $141.5_{\pm21.8}$ | $23018_{\pm1}$ |
| | SRL | 1e-02 | $.109_{\pm.028}$ | $.501_{\pm.003}$ | $.022_{\pm.032}$ | $.022_{\pm.032}$ | $142.5_{\pm23.8}$ | $21969_{\pm0}$ |
| | NG | 1e-04 | $.092_{\pm.004}$ | $.499_{\pm.001}$ | $.001_{\pm.000}$ | $.001_{\pm.000}$ | $38.3_{\pm6.1}$ | $30604_{\pm4}$ |
| | NG | 1e-03 | $.092_{\pm.004}$ | $.500_{\pm.001}$ | $.000_{\pm.000}$ | $.000_{\pm.000}$ | $37.7_{\pm5.6}$ | $29559_{\pm6}$ |
| | NG | 1e-02 | $.092_{\pm.004}$ | $.500_{\pm.001}$ | $.000_{\pm.000}$ | $.000_{\pm.000}$ | $37.8_{\pm5.8}$ | $28513_{\pm6}$ |
| | ANG | 1e-04 | $.282_{\pm.027}$ | $.536_{\pm.016}$ | $.154_{\pm.011}$ | $.180_{\pm.027}$ | $224.1_{\pm32.7}$ | $39013_{\pm8}$ |
| | ANG | 1e-03 | $.092_{\pm.004}$ | $.504_{\pm.004}$ | $.001_{\pm.001}$ | $.001_{\pm.001}$ | $217.8_{\pm21.9}$ | $37965_{\pm6}$ |
| | ANG | 1e-02 | $.091_{\pm.004}$ | $.505_{\pm.004}$ | $.000_{\pm.000}$ | $.000_{\pm.000}$ | $232.4_{\pm23.7}$ | $36917_{\pm2}$ |
| | UNSIR | 1e-04 | $.215_{\pm.014}$ | $.723_{\pm.027}$ | $.511_{\pm.008}$ | $.903_{\pm.030}$ | $311.5_{\pm18.8}$ | $47570_{\pm9}$ |
| | UNSIR | 1e-03 | $.452_{\pm.071}$ | $.652_{\pm.033}$ | $.436_{\pm.005}$ | $.692_{\pm.067}$ | $317.6_{\pm29.6}$ | $46519_{\pm10}$ |
| | UNSIR | 1e-02 | $.182_{\pm.097}$ | $.499_{\pm.004}$ | $.087_{\pm.084}$ | $.085_{\pm.083}$ | $320.8_{\pm35.1}$ | $45472_{\pm8}$ |
| SWM | CF−$k$ | 1e-04 | $.257_{\pm.036}$ | $.711_{\pm.030}$ | $.490_{\pm.007}$ | $.872_{\pm.052}$ | $84.7_{\pm16.8}$ | $22545_{\pm1}$ |
| | CF−$k$ | 1e-03 | $.255_{\pm.035}$ | $.710_{\pm.027}$ | $.490_{\pm.007}$ | $.874_{\pm.051}$ | $87.1_{\pm19.3}$ | $21980_{\pm2}$ |
| | CF−$k$ | 1e-02 | $.322_{\pm.029}$ | $.693_{\pm.024}$ | $.462_{\pm.009}$ | $.817_{\pm.042}$ | $89.9_{\pm18.7}$ | $22358_{\pm6}$ |
| | EU−$k$ | 1e-04 | $.240_{\pm.037}$ | $.709_{\pm.036}$ | $.490_{\pm.007}$ | $.866_{\pm.060}$ | $1733.0_{\pm307.7}$ | $23960_{\pm3}$ |
| | EU−$k$ | 1e-03 | $.241_{\pm.026}$ | $.707_{\pm.028}$ | $.488_{\pm.010}$ | $.868_{\pm.051}$ | $1730.0_{\pm307.1}$ | $23395_{\pm4}$ |
| | EU−$k$ | 1e-02 | $.264_{\pm.031}$ | $.697_{\pm.027}$ | $.476_{\pm.004}$ | $.845_{\pm.051}$ | $1733.0_{\pm313.3}$ | $23110_{\pm3}$ |
| | SalUn | 1e-04 | $.238_{\pm.008}$ | $.724_{\pm.028}$ | $.489_{\pm.021}$ | $.892_{\pm.030}$ | $153.4_{\pm1.7}$ | $11317_{\pm1327}$ |
| | SalUn | 1e-03 | $.460_{\pm.147}$ | $.659_{\pm.048}$ | $.434_{\pm.021}$ | $.693_{\pm.129}$ | $153.2_{\pm1.6}$ | $10081_{\pm1329}$ |
| | SalUn | 1e-02 | $.092_{\pm.004}$ | $.500_{\pm.001}$ | $.000_{\pm.000}$ | $.000_{\pm.000}$ | $153.4_{\pm1.7}$ | $8844_{\pm1327}$ |
| DIS | BT | 1e-04 | $.202_{\pm.042}$ | $.651_{\pm.013}$ | $.074_{\pm.014}$ | $.172_{\pm.054}$ | $286.7_{\pm50.2}$ | $40651_{\pm10}$ |
| | BT | 1e-03 | $.155_{\pm.017}$ | $.654_{\pm.032}$ | $.037_{\pm.010}$ | $.129_{\pm.034}$ | $282.2_{\pm43.0}$ | $39415_{\pm12}$ |
| | BT | 1e-02 | $.092_{\pm.002}$ | $.507_{\pm.003}$ | $.001_{\pm.001}$ | $.001_{\pm.002}$ | $274.4_{\pm29.3}$ | $38177_{\pm11}$ |
| | SCRUB | 1e-04 | $.339_{\pm.100}$ | $.679_{\pm.038}$ | $.460_{\pm.025}$ | $.809_{\pm.103}$ | $143.2_{\pm15.6}$ | $44366_{\pm9}$ |
| | SCRUB | 1e-03 | $.217_{\pm.088}$ | $.530_{\pm.013}$ | $.104_{\pm.067}$ | $.127_{\pm.086}$ | $145.6_{\pm19.7}$ | $43319_{\pm8}$ |
| | SCRUB | 1e-02 | $.091_{\pm.004}$ | $.499_{\pm.001}$ | $.000_{\pm.000}$ | $.000_{\pm.000}$ | $143.2_{\pm17.6}$ | $42272_{\pm8}$ |
| WI | FF | 1e-08 | $.341_{\pm.294}$ | $.065_{\pm.922}$ | $.186_{\pm.161}$ | $.282_{\pm.244}$ | $2340.0_{\pm121.8}$ | $2993_{\pm1429}$ |
| | FF | 1e-07 | $.337_{\pm.291}$ | $.071_{\pm.927}$ | $.186_{\pm.161}$ | $.290_{\pm.251}$ | $2424.0_{\pm395.7}$ | $2866_{\pm0}$ |
| | FF | 1e-06 | $.340_{\pm.293}$ | $.070_{\pm.927}$ | $.192_{\pm.166}$ | $.297_{\pm.257}$ | $2429.0_{\pm381.4}$ | $1915_{\pm0}$ |
| | SSD | 1e-04 | $.316_{\pm.066}$ | $.681_{\pm.030}$ | $.466_{\pm.007}$ | $.830_{\pm.071}$ | $120.9_{\pm.8}$ | $8118_{\pm500}$ |
| | SSD | 1e-03 | $.316_{\pm.066}$ | $.682_{\pm.030}$ | $.466_{\pm.007}$ | $.830_{\pm.071}$ | $121.1_{\pm3.1}$ | $7254_{\pm499}$ |
| | SSD | 1e-02 | $.314_{\pm.064}$ | $.682_{\pm.030}$ | $.466_{\pm.007}$ | $.830_{\pm.071}$ | $128.7_{\pm12.5}$ | $47620_{\pm12}$ |

Table 11: Comparison of Unlearners on ResNet18 trained on CelebA (mean $\pm$ std).

| Group | Method | LR | LUMA | UMIA | F1 (test) | F1 (forget) | RunTime | GPU (MB) |
|---|---|---|---|---|---|---|---|---|
| − | Orig. | - | .374 ± .095 | .642 ± .128 | .668 ± .027 | .811 ± .039 | 2202.0 ± 329.6 | 5379 ± 2173 |
| − | Gold | - | .636 ± .000 | .541 ± .007 | .589 ± .064 | .583 ± .040 | 2202.0 ± 329.6 | 5379 ± 2173 |
| FT | GD | 1e-04 | .418 ± .122 | .581 ± .102 | .697 ± .008 | .835 ± .078 | 332.0 ± 33.1 | 7225 ± 154 |
| | GD | 1e-03 | .657 ± .156 | .551 ± .078 | .607 ± .066 | .684 ± .139 | 352.0 ± 42.3 | 7018 ± 281 |
| | GD | 1e-02 | .664 ± .134 | .535 ± .005 | .572 ± .016 | .535 ± .024 | 1453.0 ± 1826.0 | 6659 ± 292 |
| | SRL | 1e-04 | .422 ± .098 | .572 ± .086 | .703 ± .008 | .835 ± .065 | 227.6 ± 33.8 | 3433 ± 407 |
| | SRL | 1e-03 | .710 ± .062 | .525 ± .016 | .638 ± .041 | .657 ± .084 | 229.4 ± 26.3 | 3063 ± 202 |
| | SRL | 1e-02 | .707 ± .118 | .529 ± .008 | .562 ± .072 | .548 ± .035 | 228.1 ± 41.4 | 2692 ± 1 |
| | NG | 1e-04 | .485 ± .154 | .605 ± .085 | .680 ± .015 | .825 ± .085 | 1.5 ± .2 | 7408 ± 148 |
| | NG | 1e-03 | .119 ± .025 | .563 ± .023 | .191 ± .043 | .194 ± .040 | 1.7 ± .1 | 7276 ± 152 |
| | NG | 1e-02 | .040 ± .015 | .529 ± .017 | .045 ± .022 | .043 ± .030 | 4.1 ± 3.8 | 6905 ± 152 |
| | ANG | 1e-04 | .079 ± .032 | .848 ± .007 | .179 ± .022 | .122 ± .012 | 1267.0 ± 861.5 | 8664 ± 151 |
| | ANG | 1e-03 | .073 ± .034 | .779 ± .017 | .127 ± .013 | .132 ± .016 | 779.5 ± 48.9 | 8526 ± 152 |
| | ANG | 1e-02 | .041 ± .013 | .546 ± .018 | .052 ± .012 | .051 ± .016 | 1889.0 ± 1935.0 | 8393 ± 151 |
| | UNSIR | 1e-04 | .398 ± .137 | .585 ± .107 | .698 ± .007 | .837 ± .076 | 1084.0 ± 1271.0 | 10262 ± 151 |
| | UNSIR | 1e-03 | .614 ± .183 | .553 ± .068 | .603 ± .032 | .683 ± .117 | 1455.0 ± 1898.0 | 10129 ± 153 |
| | UNSIR | 1e-02 | .704 ± .038 | .532 ± .008 | .580 ± .028 | .561 ± .021 | 368.7 ± 29.8 | 9996 ± 152 |
| SWM | CF−$k$ | 1e-04 | .373 ± .109 | .617 ± .127 | .701 ± .011 | .867 ± .077 | 338.8 ± 41.4 | 6742 ± 202 |
| | CF−$k$ | 1e-03 | .374 ± .109 | .621 ± .114 | .700 ± .011 | .870 ± .074 | 331.7 ± 4.5 | 6742 ± 202 |
| | CF−$k$ | 1e-02 | .371 ± .134 | .627 ± .111 | .694 ± .008 | .861 ± .078 | 1282.0 ± 1654.0 | 7361 ± 152 |
| | EU−$k$ | 1e-04 | .337 ± .086 | .618 ± .123 | .700 ± .011 | .864 ± .077 | 1817.0 ± 37.0 | 6698 ± 152 |
| | EU−$k$ | 1e-03 | .338 ± .093 | .621 ± .122 | .700 ± .011 | .866 ± .074 | 1717.0 ± 203.4 | 6697 ± 152 |
| | EU−$k$ | 1e-02 | .349 ± .097 | .617 ± .113 | .694 ± .008 | .859 ± .080 | 1723.0 ± 217.5 | 6742 ± 202 |
| | SalUn | 1e-04 | .423 ± .084 | .569 ± .085 | .703 ± .009 | .832 ± .054 | 210.8 ± 26.9 | 5704 ± 960 |
| | SalUn | 1e-03 | .713 ± .085 | .538 ± .030 | .613 ± .017 | .639 ± .059 | 216.1 ± 37.2 | 5124 ± 961 |
| | SalUn | 1e-02 | .750 ± .095 | .540 ± .008 | .613 ± .024 | .603 ± .028 | 216.8 ± 28.6 | 4545 ± 960 |
| DIS | BT | 1e-04 | .084 ± .055 | .547 ± .014 | .126 ± .113 | .141 ± .119 | 784.9 ± 71.4 | 8629 ± 193 |
| | BT | 1e-03 | .094 ± .066 | .554 ± .006 | .122 ± .118 | .147 ± .136 | 789.0 ± 64.9 | 8452 ± 193 |
| | BT | 1e-02 | .081 ± .021 | .555 ± .004 | .135 ± .085 | .150 ± .094 | 780.4 ± 82.4 | 8573 ± 149 |
| | SCRUB | 1e-04 | .399 ± .119 | .580 ± .101 | .707 ± .011 | .836 ± .075 | 442.9 ± 69.9 | 9218 ± 148 |
| | SCRUB | 1e-03 | .656 ± .008 | .545 ± .021 | .575 ± .044 | .619 ± .085 | 465.5 ± 41.2 | 9086 ± 151 |
| | SCRUB | 1e-02 | .386 ± .081 | .562 ± .008 | .401 ± .056 | .369 ± .026 | 429.7 ± 66.1 | 8951 ± 150 |
| WI | FF | 1e-08 | .659 ± .167 | .554 ± .054 | .595 ± .032 | .689 ± .072 | 76.1 ± 2.3 | 3308 ± 1305 |
| | FF | 1e-07 | .670 ± .059 | .569 ± .052 | .576 ± .029 | .645 ± .092 | 75.3 ± .6 | 2861 ± 1305 |
| | FF | 1e-06 | .679 ± .124 | .568 ± .052 | .580 ± .031 | .678 ± .070 | 75.8 ± 1.4 | 2416 ± 1306 |
| | SSD | 1e-04 | .419 ± .140 | .610 ± .117 | .679 ± .009 | .842 ± .090 | 342.7 ± 54.0 | 9277 ± 147 |
| | SSD | 1e-03 | .400 ± .158 | .614 ± .120 | .679 ± .009 | .842 ± .090 | 127.0 ± 1634.0 | 9232 ± 147 |
| | SSD | 1e-02 | .407 ± .147 | .612 ± .125 | .679 ± .009 | .842 ± .090 | 685.5 ± 613.3 | 9291 ± 146 |

Table 12: Comparison of Unlearners on ResNet50 trained on CelebA (mean ± std).

| Group | Method | LR | LUMA | UMIA | F1 (test) | F1 (forget) | RunTime | GPU (MB) |
|---|---|---|---|---|---|---|---|---|
| − | Orig. | - | .303 ± .027 | .770 ± .011 | .688 ± .009 | .962 ± .016 | 8967.0 ± 744.0 | 15814 ± 13 |
| − | Gold | - | .636 ± .000 | .539 ± .008 | .670 ± .009 | .633 ± .016 | 8967.0 ± 744.0 | 15814 ± 13 |
| FT | GD | 1e-04 | .367 ± .012 | .748 ± .011 | .704 ± .003 | .943 ± .007 | 284.2 ± 20.6 | 19048 ± 6 |
| | GD | 1e-03 | .456 ± .032 | .703 ± .013 | .680 ± .004 | .892 ± .006 | 294.4 ± 37.8 | 18003 ± 8 |
| | GD | 1e-02 | .673 ± .117 | .535 ± .006 | .578 ± .039 | .544 ± .032 | 291.3 ± 31.8 | 16955 ± 11 |
| | SRL | 1e-04 | .388 ± .018 | .729 ± .010 | .704 ± .003 | .929 ± .012 | 401.1 ± 22.4 | 17758 ± 7660 |
| | SRL | 1e-03 | .621 ± .041 | .623 ± .028 | .665 ± .011 | .784 ± .033 | 302.6 ± 50.1 | 16709 ± 7664 |
| | SRL | 1e-02 | .734 ± .011 | .529 ± .005 | .607 ± .011 | .564 ± .016 | 291.8 ± 32.7 | 15662 ± 7669 |
| | NG | 1e-04 | .397 ± .023 | .733 ± .006 | .687 ± .004 | .953 ± .003 | 1.5 ± .1 | 27943 ± 1325 |
| | NG | 1e-03 | .314 ± .078 | .545 ± .015 | .376 ± .054 | .421 ± .055 | 1.6 ± .1 | 26896 ± 1323 |
| | NG | 1e-02 | .041 ± .015 | .556 ± .011 | .107 ± .063 | .118 ± .066 | 1.5 ± .0 | 25850 ± 1320 |
| | ANG | 1e-04 | .054 ± .025 | .878 ± .007 | .168 ± .063 | .147 ± .048 | 714.4 ± 54.1 | 35356 ± 1329 |
| | ANG | 1e-03 | .071 ± .019 | .750 ± .099 | .216 ± .055 | .174 ± .006 | 693.4 ± 26.1 | 34304 ± 1328 |
| | ANG | 1e-02 | .045 ± .018 | .558 ± .013 | .126 ± .058 | .116 ± .048 | 689.8 ± 16.6 | 33257 ± 1328 |
| | UNSIR | 1e-04 | .369 ± .012 | .744 ± .005 | .705 ± .003 | .944 ± .008 | 296.6 ± 17.4 | 35105 ± 12965 |
| | UNSIR | 1e-03 | .460 ± .041 | .703 ± .008 | .669 ± .026 | .878 ± .058 | 295.2 ± 18.1 | 34058 ± 12970 |
| | UNSIR | 1e-02 | .704 ± .091 | .529 ± .008 | .589 ± .044 | .564 ± .061 | 294.1 ± 19.2 | 33000 ± 12969 |
| SWM | CF−$k$ | 1e-04 | .332 ± .019 | .784 ± .009 | .699 ± .002 | .969 ± .001 | 248.1 ± 20.6 | 20192 ± 1153 |
| | CF−$k$ | 1e-03 | .335 ± .016 | .780 ± .005 | .698 ± .002 | .968 ± .001 | 246.4 ± 17.5 | 19629 ± 1153 |
| | CF−$k$ | 1e-02 | .340 ± .024 | .772 ± .008 | .697 ± .001 | .969 ± .004 | 250.1 ± 24.6 | 19913 ± 1658 |
| | EU−$k$ | 1e-04 | .325 ± .016 | .775 ± .003 | .698 ± .001 | .968 ± .001 | 1284.0 ± 68.4 | 22261 ± 1149 |
| | EU−$k$ | 1e-03 | .329 ± .013 | .768 ± .006 | .697 ± .002 | .968 ± .001 | 1269.0 ± 78.4 | 21696 ± 1149 |
| | EU−$k$ | 1e-02 | .323 ± .012 | .780 ± .008 | .697 ± .002 | .968 ± .002 | 1234.0 ± 93.1 | 20756 ± 1152 |
| | SalUn | 1e-04 | .379 ± .013 | .736 ± .008 | .704 ± .003 | .932 ± .010 | 486.0 ± 167.0 | 15095 ± 8763 |
| | SalUn | 1e-03 | .600 ± .059 | .620 ± .020 | .682 ± .005 | .806 ± .023 | 430.3 ± 199.6 | 27398 ± 18090 |
| | SalUn | 1e-02 | .586 ± .035 | .529 ± .008 | .548 ± .015 | .511 ± .027 | 419.5 ± 216.1 | 26158 ± 18089 |
| DIS | BT | 1e-04 | .340 ± .128 | .618 ± .021 | .318 ± .091 | .607 ± .150 | 744.4 ± 69.6 | 30182 ± 12965 |
| | BT | 1e-03 | .027 ± .006 | .567 ± .033 | .024 ± .015 | .102 ± .082 | 735.1 ± 66.2 | 28946 ± 12968 |
| | BT | 1e-02 | .030 ± .007 | .547 ± .006 | .074 ± .036 | .081 ± .047 | 711.2 ± 30.0 | 27715 ± 12965 |
| | SCRUB | 1e-04 | .369 ± .013 | .751 ± .005 | .697 ± .006 | .942 ± .010 | 371.8 ± 41.9 | 33746 ± 12968 |
| | SCRUB | 1e-03 | .570 ± .019 | .638 ± .008 | .648 ± .016 | .808 ± .033 | 418.9 ± 126.4 | 32701 ± 12970 |
| | SCRUB | 1e-02 | .234 ± .040 | .551 ± .024 | .365 ± .016 | .352 ± .029 | 371.7 ± 46.2 | 31655 ± 12969 |
| WI | FF | 1e-08 | .547 ± .042 | .672 ± .023 | .625 ± .045 | .793 ± .078 | 211.2 ± 30.2 | 3798 ± 0 |
| | FF | 1e-07 | .575 ± .071 | .655 ± .024 | .617 ± .033 | .775 ± .067 | 205.6 ± 26.5 | 2841 ± 0 |
| | FF | 1e-06 | .550 ± .041 | .659 ± .020 | .629 ± .012 | .811 ± .029 | 196.7 ± 30.7 | 1885 ± 0 |
| | SSD | 1e-04 | .350 ± .040 | .773 ± .013 | .688 ± .009 | .962 ± .017 | 280.2 ± 40.3 | 36018 ± 12967 |
| | SSD | 1e-03 | .346 ± .045 | .780 ± .022 | .688 ± .009 | .962 ± .017 | 281.9 ± 43.9 | 35161 ± 12967 |
| | SSD | 1e-02 | .348 ± .044 | .777 ± .021 | .688 ± .009 | .962 ± .017 | 273.8 ± 28.8 | 34306 ± 12968 |

Table 13: Comparison of Unlearners on DistilBERT trained on AG News (mean ± std).

| Group | Method | LR | LUMA | UMIA | F1 (test) | F1 (forget) | RunTime | GPU (MB) |
|---|---|---|---|---|---|---|---|---|
| − | Orig. | - | .531 ± .004 | .574 ± .005 | .935 ± .002 | .962 ± .003 | 1629.0 ± 4.1 | 7171 ± 889 |
| − | Gold | - | .636 ± .000 | .542 ± .004 | .934 ± .001 | .822 ± .005 | 1629.0 ± 4.1 | 7171 ± 889 |
| FT | GD | 1e-06 | .632 ± .019 | .559 ± .004 | .935 ± .001 | .956 ± .009 | 541.5 ± 5.1 | 13081 ± 0 |
| | GD | 1e-05 | .620 ± .006 | .563 ± .003 | .931 ± .002 | .959 ± .004 | 537.4 ± 1.5 | 33827 ± 0 |
| | GD | 1e-04 | .676 ± .064 | .558 ± .007 | .923 ± .006 | .760 ± .076 | 537.8 ± 1.7 | 33062 ± 0 |
| | SRL | 1e-06 | .682 ± .026 | .533 ± .009 | .936 ± .001 | .880 ± .070 | 563.9 ± 5.0 | 13843 ± 0 |
| | SRL | 1e-05 | .662 ± .026 | .544 ± .006 | .929 ± .003 | .921 ± .038 | 558.2 ± .9 | 36124 ± 0 |
| | SRL | 1e-04 | .694 ± .034 | .560 ± .013 | .924 ± .002 | .792 ± .057 | 557.2 ± 1.3 | 35357 ± 0 |
| | NG | 1e-06 | .457 ± .305 | .669 ± .177 | .830 ± .112 | .574 ± .156 | 20.7 ± .4 | 14608 ± 0 |
| | NG | 1e-05 | .014 ± .005 | .857 ± .001 | .269 ± .011 | .140 ± .080 | 21.0 ± .0 | 38420 ± 0 |
| | NG | 1e-04 | .004 ± .000 | .857 ± .000 | .100 ± .000 | .011 ± .000 | 21.0 ± .0 | 37654 ± 0 |
| | ANG | 1e-06 | .153 ± .022 | .975 ± .002 | .918 ± .001 | .304 ± .084 | 1087.0 ± 11.5 | 17788 ± 0 |
| | ANG | 1e-05 | .079 ± .001 | .988 ± .000 | .926 ± .005 | .014 ± .006 | 1071.0 ± 4.3 | 20877 ± 0 |
| | ANG | 1e-04 | .025 ± .034 | .971 ± .026 | .412 ± .393 | .008 ± .003 | 1088.0 ± 3.5 | 42367 ± 0 |
| | UNSIR | 1e-06 | .627 ± .017 | .560 ± .003 | .935 ± .001 | .954 ± .007 | 584.1 ± 6.5 | 16116 ± 4429 |
| | UNSIR | 1e-05 | .617 ± .007 | .561 ± .002 | .931 ± .001 | .961 ± .005 | 579.1 ± 1.5 | 23294 ± 0 |
| | UNSIR | 1e-04 | .683 ± .034 | .559 ± .012 | .917 ± .006 | .782 ± .060 | 579.3 ± 1.7 | 22526 ± 0 |
| SWM | CF−$k$ | 1e-06 | .670 ± .014 | .574 ± .003 | .935 ± .002 | .962 ± .003 | 211.7 ± 6.2 | 13089 ± 0 |
| | CF−$k$ | 1e-05 | .675 ± .022 | .571 ± .003 | .935 ± .002 | .957 ± .011 | 205.0 ± 1.0 | 34862 ± 0 |
| | CF−$k$ | 1e-04 | .690 ± .025 | .561 ± .004 | .935 ± .002 | .951 ± .012 | 205.5 ± 1.1 | 34862 ± 0 |
| | EU−$k$ | 1e-06 | .458 ± .052 | .748 ± .047 | .932 ± .004 | .955 ± .020 | 636.5 ± 19.9 | 12581 ± 0 |
| | EU−$k$ | 1e-05 | .612 ± .019 | .568 ± .001 | .935 ± .002 | .955 ± .013 | 615.3 ± 2.1 | 34862 ± 0 |
| | EU−$k$ | 1e-04 | .626 ± .019 | .557 ± .005 | .935 ± .002 | .949 ± .010 | 616.2 ± 1.8 | 34862 ± 0 |
| | SalUn | 1e-06 | .659 ± .028 | .534 ± .006 | .935 ± .001 | .880 ± .070 | 736.2 ± 14.9 | 17408 ± 4430 |
| | SalUn | 1e-05 | .619 ± .032 | .548 ± .007 | .931 ± .001 | .940 ± .038 | 764.2 ± 23.5 | 32588 ± 590 |
| | SalUn | 1e-04 | .634 ± .041 | .549 ± .007 | .926 ± .007 | .733 ± .045 | 767.6 ± 13.1 | 31312 ± 590 |
| DIS | BT | 1e-06 | .080 ± .047 | .898 ± .014 | .401 ± .142 | .481 ± .072 | 985.6 ± 14.2 | 14990 ± 4430 |
| | BT | 1e-05 | .043 ± .011 | .975 ± .003 | .123 ± .040 | .611 ± .036 | 972.5 ± 2.7 | 23190 ± 0 |
| | BT | 1e-04 | .013 ± .006 | .912 ± .095 | .109 ± .024 | .292 ± .083 | 971.2 ± 2.8 | 22170 ± 0 |
| | SCRUB | 1e-06 | .608 ± .014 | .561 ± .003 | .936 ± .001 | .956 ± .003 | 731.7 ± 6.4 | 15457 ± 4430 |
| | SCRUB | 1e-05 | .598 ± .006 | .563 ± .003 | .934 ± .001 | .963 ± .002 | 726.8 ± 2.0 | 25186 ± 0 |
| | SCRUB | 1e-04 | .581 ± .082 | .573 ± .055 | .869 ± .067 | .746 ± .014 | 726.7 ± 1.8 | 24422 ± 0 |
| WI | FF | 1e-08 | .010 ± .004 | .709 ± .105 | .179 ± .039 | .135 ± .112 | .1 ± .0 | 7220 ± 0 |
| | FF | 1e-07 | .008 ± .004 | .699 ± .029 | .147 ± .028 | .111 ± .111 | .1 ± .0 | 6195 ± 0 |
| | FF | 1e-06 | .005 ± .001 | .711 ± .158 | .136 ± .032 | .012 ± .001 | 1.0 ± 1.3 | 5169 ± 0 |
| | SSD | 1e-06 | .612 ± .012 | .577 ± .003 | .935 ± .002 | .962 ± .003 | 553.4 ± 4.9 | 16072 ± 4430 |
| | SSD | 1e-05 | .607 ± .007 | .580 ± .004 | .935 ± .002 | .962 ± .003 | 547.7 ± 1.4 | 29382 ± 0 |
| | SSD | 1e-04 | .611 ± .008 | .575 ± .003 | .935 ± .002 | .962 ± .003 | 548.9 ± 2.0 | 29127 ± 0 |

Table 14: Comparison of Unlearners on BERT trained on AG News (mean ± std).

| Group | Method | LR | LUMA | UMIA | F1 (test) | F1 (forget) | RunTime | GPU (MB) |
|---|---|---|---|---|---|---|---|---|
| − | Orig. | - | .399 ± .056 | .641 ± .030 | .880 ± .012 | .916 ± .008 | 520.0 ± 54.7 | 56962 ± 27786 |
| − | Gold | - | .636 ± .000 | .525 ± .024 | .906 ± .005 | .666 ± .036 | 520.0 ± 54.7 | 56962 ± 27786 |
| FT | GD | 1e-06 | .543 ± .069 | .559 ± .007 | .913 ± .001 | .876 ± .017 | 1685.0 ± 1.7 | 32123 ± 0 |
| | GD | 1e-05 | .544 ± .063 | .568 ± .009 | .908 ± .003 | .867 ± .019 | 1655.0 ± 24.9 | 75876 ± 2154 |
| | GD | 1e-04 | .235 ± .377 | .645 ± .162 | .436 ± .412 | .236 ± .318 | 1655.0 ± 25.0 | 75473 ± 2819 |
| | SRL | 1e-06 | .554 ± .061 | .545 ± .006 | .913 ± .001 | .866 ± .018 | 1748.0 ± .4 | 25943 ± 0 |
| | SRL | 1e-05 | .532 ± .000 | .567 ± .000 | .909 ± .000 | .872 ± .000 | 1702.0 ± .0 | 77502 ± 0 |
| | SRL | 1e-04 | .008 ± .000 | .631 ± .000 | .103 ± .000 | .001 ± .000 | 1702.0 ± .0 | 76249 ± 0 |
| | NG | 1e-06 | .158 ± .028 | .852 ± .000 | .570 ± .004 | .369 ± .040 | 65.3 ± .1 | 27199 ± 0 |
| | NG | 1e-05 | .012 ± .004 | .859 ± .001 | .185 ± .061 | .011 ± .000 | 65.6 ± .1 | 44945 ± 6656 |
| | NG | 1e-04 | .009 ± .001 | .859 ± .001 | .100 ± .000 | .011 ± .000 | 65.6 ± .0 | 43691 ± 6657 |
| | ANG | 1e-06 | .185 ± .006 | .975 ± .001 | .893 ± .002 | .316 ± .010 | 3414.0 ± 8.4 | 33283 ± 0 |
| | ANG | 1e-05 | .105 ± .006 | .986 ± .001 | .867 ± .006 | .011 ± .000 | 3413.0 ± 2.4 | 50323 ± 3865 |
| | ANG | 1e-04 | .009 ± .001 | .635 ± .035 | .105 ± .005 | .007 ± .006 | 3413.0 ± 2.3 | 49065 ± 3865 |
| | UNSIR | 1e-06 | .543 ± .072 | .558 ± .009 | .913 ± .000 | .871 ± .023 | 1844.0 ± 51.3 | 34828 ± 0 |
| | UNSIR | 1e-05 | .575 ± .095 | .554 ± .015 | .910 ± .002 | .832 ± .050 | 1814.0 ± .7 | 54186 ± 3700 |
| | UNSIR | 1e-04 | .022 ± .022 | .828 ± .026 | .187 ± .145 | .080 ± .137 | 1818.0 ± 6.3 | 52931 ± 3700 |
| SWM | CF−$k$ | 1e-06 | .492 ± .074 | .646 ± .013 | .893 ± .005 | .902 ± .027 | 596.3 ± .4 | 23755 ± 0 |
| | CF−$k$ | 1e-05 | .543 ± .065 | .580 ± .014 | .897 ± .011 | .895 ± .016 | 595.7 ± 5.6 | 74400 ± 4542 |
| | CF−$k$ | 1e-04 | .586 ± .076 | .555 ± .017 | .905 ± .005 | .873 ± .013 | 599.1 ± 5.9 | 74400 ± 4542 |
| | EU−$k$ | 1e-06 | .421 ± .101 | .723 ± .051 | .889 ± .006 | .879 ± .060 | 1789.0 ± .6 | 22922 ± 0 |
| | EU−$k$ | 1e-05 | .510 ± .059 | .587 ± .023 | .898 ± .008 | .882 ± .036 | 1784.0 ± 16.7 | 74400 ± 4542 |
| | EU−$k$ | 1e-04 | .538 ± .059 | .556 ± .014 | .907 ± .004 | .876 ± .007 | 1785.0 ± 17.2 | 74400 ± 4542 |
| | SalUn | 1e-06 | .543 ± .057 | .548 ± .007 | .913 ± .000 | .864 ± .016 | 218.0 ± 73.0 | 35988 ± 0 |
| | SalUn | 1e-05 | .556 ± .047 | .552 ± .006 | .912 ± .001 | .846 ± .002 | 2154.0 ± 1.8 | 69079 ± 6163 |
| | SalUn | 1e-04 | .008 ± .001 | .827 ± .020 | .102 ± .001 | .007 ± .005 | 2152.0 ± 7.7 | 66978 ± 6165 |
| DIS | BT | 1e-06 | .083 ± .027 | .784 ± .140 | .289 ± .044 | .421 ± .088 | 299.0 ± 78.3 | 32106 ± 0 |
| | BT | 1e-05 | .041 ± .009 | .962 ± .010 | .104 ± .002 | .433 ± .077 | 2942.0 ± 2.9 | 57465 ± 5999 |
| | BT | 1e-04 | .017 ± .009 | .770 ± .079 | .097 ± .004 | .166 ± .136 | 2944.0 ± 4.9 | 55790 ± 5998 |
| | SCRUB | 1e-06 | .533 ± .066 | .556 ± .008 | .912 ± .000 | .871 ± .022 | 2287.0 ± 61.6 | 32623 ± 0 |
| | SCRUB | 1e-05 | .594 ± .087 | .563 ± .003 | .888 ± .011 | .779 ± .073 | 2258.0 ± 13.4 | 61258 ± 6666 |
| | SCRUB | 1e-04 | .012 ± .007 | .803 ± .094 | .160 ± .099 | .004 ± .004 | 2258.0 ± 13.1 | 60007 ± 6666 |
| WI | FF | 1e-08 | .009 ± .001 | .854 ± .007 | .142 ± .058 | .011 ± .001 | .3 ± .0 | 76257 ± 2091 |
| | FF | 1e-07 | .009 ± .002 | .759 ± .114 | .111 ± .012 | .020 ± .027 | .3 ± .0 | 75371 ± 3213 |
| | FF | 1e-06 | .008 ± .001 | .840 ± .020 | .101 ± .001 | .010 ± .001 | 2.3 ± .5 | 78113 ± 4 |
| | SSD | 1e-06 | .440 ± .065 | .654 ± .012 | .879 ± .010 | .907 ± .021 | 1793.0 ± 46.3 | 33729 ± 0 |
| | SSD | 1e-05 | .440 ± .064 | .652 ± .008 | .879 ± .010 | .907 ± .021 | 1765.0 ± 1.1 | 58845 ± 2894 |
| | SSD | 1e-04 | .439 ± .063 | .653 ± .010 | .879 ± .010 | .907 ± .021 | 1765.0 ± 1.0 | 58428 ± 2894 |

Table 15: Comparison of Unlearners on BERT trained on DistilBERT (mean $\pm$ std).

| Group | Method | LR | LUMA | UMIA | F1 (test) | F1 (forget) | RunTime | GPU (MB) |
|---|---|---|---|---|---|---|---|---|
| − | Orig. | - | $.564 \pm .011$ | $.548 \pm .002$ | $.933 \pm .003$ | $.981 \pm .006$ | $949.0 \pm 7.8$ | $32839 \pm 0$ |
| − | Gold | - | $.636 \pm .000$ | $.500 \pm .004$ | $.925 \pm .005$ | $.920 \pm .009$ | $949.0 \pm 7.8$ | $32839 \pm 0$ |
| FT | GD | 1e-06 | $.632 \pm .013$ | $.544 \pm .004$ | $.939 \pm .001$ | $.988 \pm .002$ | $464.9 \pm 3.3$ | $38165 \pm 0$ |
| | GD | 1e-05 | $.640 \pm .015$ | $.545 \pm .006$ | $.934 \pm .002$ | $.983 \pm .002$ | $465.1 \pm 3.1$ | $34185 \pm 0$ |
| | GD | 1e-04 | $.685 \pm .009$ | $.512 \pm .006$ | $.899 \pm .008$ | $.899 \pm .008$ | $465.0 \pm 3.0$ | $33383 \pm 0$ |
| | SRL | 1e-06 | $.630 \pm .012$ | $.545 \pm .002$ | $.938 \pm .001$ | $.986 \pm .002$ | $490.9 \pm 3.2$ | $34978 \pm 0$ |
| | SRL | 1e-05 | $.632 \pm .017$ | $.544 \pm .003$ | $.936 \pm .003$ | $.984 \pm .004$ | $491.3 \pm 3.1$ | $36570 \pm 0$ |
| | SRL | 1e-04 | $.241 \pm .374$ | $.508 \pm .016$ | $.524 \pm .328$ | $.531 \pm .343$ | $490.2 \pm 3.0$ | $35775 \pm 0$ |
| | NG | 1e-06 | $.301 \pm .448$ | $.519 \pm .031$ | $.568 \pm .308$ | $.586 \pm .334$ | $26.0 \pm .4$ | $37371 \pm 0$ |
| | NG | 1e-05 | $.027 \pm .002$ | $.502 \pm .005$ | $.334 \pm .002$ | $.337 \pm .003$ | $26.0 \pm .4$ | $38962 \pm 0$ |
| | NG | 1e-04 | $.027 \pm .002$ | $.501 \pm .004$ | $.334 \pm .002$ | $.334 \pm .006$ | $25.9 \pm .4$ | $38168 \pm 0$ |
| | ANG | 1e-06 | $.600 \pm .011$ | $.528 \pm .002$ | $.919 \pm .002$ | $.895 \pm .019$ | $932.5 \pm 7.3$ | $40966 \pm 0$ |
| | ANG | 1e-05 | $.316 \pm .109$ | $.511 \pm .010$ | $.828 \pm .081$ | $.573 \pm .073$ | $928.4 \pm 5.8$ | $42569 \pm 0$ |
| | ANG | 1e-04 | $.026 \pm .001$ | $.509 \pm .009$ | $.332 \pm .000$ | $.331 \pm .006$ | $929.2 \pm 5.6$ | $41768 \pm 0$ |
| | UNSIR | 1e-06 | $.628 \pm .011$ | $.544 \pm .003$ | $.938 \pm .001$ | $.987 \pm .001$ | $519.3 \pm 5.0$ | $18614 \pm 0$ |
| | UNSIR | 1e-05 | $.635 \pm .014$ | $.540 \pm .004$ | $.936 \pm .002$ | $.984 \pm .003$ | $520.9 \pm 4.8$ | $20215 \pm 1$ |
| | UNSIR | 1e-04 | $.680 \pm .013$ | $.518 \pm .005$ | $.899 \pm .003$ | $.909 \pm .007$ | $521.6 \pm 5.3$ | $19415 \pm 0$ |
| SWM | $CF-k$ | 1e-06 | $.706 \pm .019$ | $.553 \pm .002$ | $.934 \pm .002$ | $.982 \pm .006$ | $182.3 \pm 2.3$ | $36989 \pm 0$ |
| | $CF-k$ | 1e-05 | $.707 \pm .017$ | $.550 \pm .004$ | $.935 \pm .001$ | $.983 \pm .004$ | $183.2 \pm 2.2$ | $36467 \pm 0$ |
| | $CF-k$ | 1e-04 | $.706 \pm .014$ | $.549 \pm .001$ | $.935 \pm .002$ | $.984 \pm .003$ | $182.9 \pm 2.3$ | $36467 \pm 0$ |
| | $EU-k$ | 1e-06 | $.663 \pm .024$ | $.545 \pm .010$ | $.934 \pm .003$ | $.981 \pm .006$ | $365.2 \pm 4.4$ | $36467 \pm 0$ |
| | $EU-k$ | 1e-05 | $.655 \pm .013$ | $.550 \pm .004$ | $.935 \pm .001$ | $.983 \pm .003$ | $365.3 \pm 4.8$ | $36468 \pm 0$ |
| | $EU-k$ | 1e-04 | $.655 \pm .014$ | $.550 \pm .004$ | $.935 \pm .001$ | $.984 \pm .003$ | $365.0 \pm 4.6$ | $36467 \pm 0$ |
| | SalUn | 1e-06 | $.604 \pm .011$ | $.545 \pm .002$ | $.938 \pm .001$ | $.985 \pm .002$ | $696.8 \pm 37.9$ | $14472 \pm 14372$ |
| | SalUn | 1e-05 | $.604 \pm .006$ | $.542 \pm .006$ | $.933 \pm .003$ | $.982 \pm .002$ | $692.7 \pm 9.5$ | $32989 \pm 600$ |
| | SalUn | 1e-04 | $.383 \pm .310$ | $.505 \pm .003$ | $.669 \pm .289$ | $.688 \pm .296$ | $718.3 \pm 2.6$ | $31682 \pm 601$ |
| DIS | BT | 1e-06 | $.477 \pm .227$ | $.508 \pm .009$ | $.791 \pm .145$ | $.825 \pm .156$ | $879.3 \pm 8.7$ | $20328 \pm 0$ |
| | BT | 1e-05 | $.298 \pm .274$ | $.506 \pm .011$ | $.632 \pm .247$ | $.716 \pm .247$ | $880.4 \pm 8.6$ | $22442 \pm 7$ |
| | BT | 1e-04 | $.050 \pm .023$ | $.504 \pm .002$ | $.384 \pm .047$ | $.432 \pm .095$ | $876.9 \pm 9.6$ | $21387 \pm 5$ |
| | SCRUB | 1e-06 | $.606 \pm .012$ | $.547 \pm .002$ | $.938 \pm .000$ | $.987 \pm .002$ | $63.2 \pm 6.0$ | $22815 \pm 1$ |
| | SCRUB | 1e-05 | $.621 \pm .008$ | $.544 \pm .001$ | $.926 \pm .011$ | $.975 \pm .012$ | $63.5 \pm 6.2$ | $24405 \pm 4$ |
| | SCRUB | 1e-04 | $.658 \pm .019$ | $.512 \pm .006$ | $.895 \pm .009$ | $.908 \pm .005$ | $63.9 \pm 6.1$ | $23608 \pm 3$ |
| WI | FF | 1e-08 | $.049 \pm .031$ | $.506 \pm .006$ | $.400 \pm .079$ | $.395 \pm .065$ | $.1 \pm .0$ | $7220 \pm 0$ |
| | FF | 1e-07 | $.034 \pm .010$ | $.505 \pm .001$ | $.363 \pm .036$ | $.364 \pm .035$ | $.1 \pm .0$ | $6195 \pm 0$ |
| | FF | 1e-06 | $.027 \pm .001$ | $.503 \pm .010$ | $.337 \pm .003$ | $.337 \pm .007$ | $1.3 \pm 1.0$ | $5169 \pm 0$ |
| | SSD | 1e-06 | $.630 \pm .015$ | $.555 \pm .005$ | $.933 \pm .003$ | $.982 \pm .005$ | $479.0 \pm 4.0$ | $28196 \pm 4$ |
| | SSD | 1e-05 | $.632 \pm .015$ | $.553 \pm .005$ | $.933 \pm .003$ | $.982 \pm .005$ | $476.8 \pm 4.3$ | $28728 \pm 6$ |
| | SSD | 1e-04 | $.633 \pm .015$ | $.552 \pm .004$ | $.933 \pm .003$ | $.982 \pm .005$ | $477.4 \pm 4.5$ | $28462 \pm 5$ |

Table 16: Comparison of Unlearners on BERT trained on IMDB (mean ± std).

| Group | Method | LR | LUMA | UMIA | F1 (test) | F1 (forget) | RunTime | GPU (MB) |
|---|---|---|---|---|---|---|---|---|
| − | Orig. | - | .565 ± .006 | .549 ± .004 | .937 ± .005 | .997 ± .002 | 4914.0 ± 2965.0 | 43256 ± 3131 |
| − | Gold | - | .636 ± .000 | .501 ± .000 | .939 ± .006 | .928 ± .008 | 4914.0 ± 2965.0 | 43256 ± 3131 |
| FT | GD | 1e-06 | .658 ± .050 | .545 ± .002 | .945 ± .000 | .999 ± .001 | 1329.0 ± 2.9 | 46742 ± 6815 |
| | GD | 1e-05 | .667 ± .045 | .539 ± .001 | .941 ± .002 | .996 ± .004 | 1329.0 ± 4.1 | 52408 ± 4716 |
| | SRL | 1e-06 | .653 ± .047 | .545 ± .002 | .946 ± .000 | .999 ± .001 | 1401.0 ± .9 | 53724 ± 4712 |
| | SRL | 1e-05 | .661 ± .043 | .542 ± .003 | .939 ± .000 | .994 ± .002 | 1402.0 ± .2 | 55041 ± 4713 |
| | NG | 1e-06 | .051 ± .031 | .497 ± .000 | .408 ± .090 | .421 ± .096 | 75.9 ± 4.8 | 48925 ± 6893 |
| | NG | 1e-05 | .025 ± .002 | .501 ± .004 | .335 ± .000 | .340 ± .009 | 75.8 ± 5.0 | 57670 ± 4710 |
| | ANG | 1e-06 | .660 ± .056 | .502 ± .010 | .935 ± .003 | .953 ± .014 | 271.0 ± 8.4 | 58977 ± 4704 |
| | ANG | 1e-05 | .095 ± .101 | .527 ± .040 | .514 ± .257 | .415 ± .134 | 271.0 ± 8.5 | 60357 ± 4709 |
| | UNSIR | 1e-06 | .649 ± .049 | .545 ± .000 | .945 ± .000 | .998 ± .002 | 1478.0 ± 5.9 | 61675 ± 4702 |
| | UNSIR | 1e-05 | .656 ± .036 | .538 ± .004 | .933 ± .010 | .991 ± .002 | 1478.0 ± 6.1 | 62993 ± 4700 |
| SWM | CF−$k$ | 1e-06 | .716 ± .038 | .549 ± .006 | .938 ± .005 | .997 ± .002 | 512.7 ± 2.9 | 56362 ± 4719 |
| | CF−$k$ | 1e-05 | .715 ± .039 | .551 ± .007 | .939 ± .005 | .997 ± .001 | 512.4 ± 2.7 | 49558 ± 5998 |
| | EU−$k$ | 1e-06 | .671 ± .037 | .554 ± .002 | .937 ± .005 | .997 ± .001 | 1024.0 ± 3.6 | 49558 ± 5998 |
| | EU−$k$ | 1e-05 | .674 ± .042 | .551 ± .002 | .939 ± .004 | .997 ± .001 | 1025.0 ± 2.5 | 48925 ± 6893 |
| | SalUn | 1e-06 | .580 ± .000 | .556 ± .000 | .944 ± .000 | 1.000 ± .000 | 1983.0 ± .0 | 49122 ± 0 |
| | SalUn | 1e-05 | .596 ± .000 | .540 ± .000 | .926 ± .000 | .992 ± .000 | 1988.0 ± .0 | 61327 ± 0 |
| | SalUn | 1e-04 | .026 ± .000 | .498 ± .000 | .335 ± .000 | .344 ± .000 | 1988.0 ± .0 | 59980 ± 0 |
| DIS | BT | 1e-06 | .656 ± .000 | .540 ± .000 | .890 ± .000 | .948 ± .000 | 2427.0 ± .0 | 67629 ± 0 |
| | BT | 1e-05 | .309 ± .335 | .535 ± .017 | .672 ± .305 | .748 ± .340 | 2553.0 ± 177.3 | 53771 ± 21984 |
| | SCRUB | 1e-06 | .628 ± .064 | .548 ± .007 | .944 ± .000 | 1.000 ± .000 | 1904.0 ± 133.6 | 58627 ± 17586 |
| | SCRUB | 1e-05 | .638 ± .055 | .543 ± .001 | .935 ± .009 | .993 ± .007 | 1904.0 ± 132.5 | 60541 ± 16721 |
| | SCRUB | 1e-04 | .024 ± .000 | .500 ± .000 | .332 ± .000 | .323 ± .000 | 1998.0 ± .0 | 48219 ± 0 |
| WI | FF | 1e-08 | .065 ± .047 | .505 ± .010 | .447 ± .097 | .436 ± .097 | 91.3 ± 2.3 | 57216 ± 16119 |
| | FF | 1e-07 | .086 ± .000 | .498 ± .002 | .500 ± .011 | .497 ± .003 | 89.2 ± 3.3 | 55539 ± 16120 |
| | FF | 1e-06 | .051 ± .025 | .498 ± .003 | .423 ± .063 | .416 ± .058 | 91.1 ± 4.1 | 61832 ± 16759 |
| | SSD | 1e-06 | .642 ± .065 | .555 ± .015 | .941 ± .000 | .999 ± .001 | 1492.0 ± 83.8 | 58894 ± 16117 |
| | SSD | 1e-05 | .641 ± .063 | .556 ± .013 | .941 ± .000 | .999 ± .001 | 1492.0 ± 83.1 | 61221 ± 17715 |
| | SSD | 1e-04 | .642 ± .062 | .555 ± .012 | .941 ± .000 | .999 ± .001 | 1491.0 ± 83.7 | 60790 ± 17710 |

Table 17: Comparison of Unlearners on $MLP_{small}$ trained on Adult (mean $\pm$ std).

| Group | Method | LR | LUMA | UMIA | F1 (test) | F1 (forget) | RunTime | GPU (MB) |
|---|---|---|---|---|---|---|---|---|
| − | Orig. | - | .632 ± .002 | .498 ± .003 | .794 ± .002 | .795 ± .001 | 207.5 ± 9.1 | 21 ± 0 |
| − | Gold | - | .636 ± .000 | .499 ± .001 | .794 ± .002 | .793 ± .001 | 207.5 ± 9.1 | 21 ± 0 |
| FT | GD | 1e-04 | .699 ± .001 | .499 ± .002 | .793 ± .001 | .794 ± .001 | 140.4 ± 7.0 | 22 ± 0 |
| | GD | 1e-03 | .702 ± .010 | .498 ± .001 | .793 ± .001 | .796 ± .002 | 138.5 ± 2.5 | 22 ± 0 |
| | GD | 1e-02 | .709 ± .004 | .499 ± .000 | .787 ± .001 | .788 ± .002 | 129.2 ± 6.5 | 22 ± 0 |
| | SRL | 1e-04 | .669 ± .005 | .499 ± .001 | .788 ± .002 | .788 ± .000 | 163.8 ± 11.9 | 23 ± 0 |
| | SRL | 1e-03 | .657 ± .004 | .499 ± .001 | .792 ± .001 | .795 ± .001 | 179.4 ± 2.6 | 23 ± 0 |
| | SRL | 1e-02 | .668 ± .003 | .500 ± .002 | .790 ± .000 | .790 ± .002 | 166.1 ± 3.7 | 23 ± 0 |
| | NG | 1e-04 | .753 ± .038 | .500 ± .001 | .729 ± .016 | .732 ± .015 | 23.4 ± 1.1 | 25 ± 0 |
| | NG | 1e-03 | .107 ± .072 | .501 ± .002 | .352 ± .141 | .352 ± .137 | 22.8 ± .3 | 25 ± 0 |
| | NG | 1e-02 | .148 ± .001 | .498 ± .002 | .434 ± .000 | .432 ± .000 | 25.5 ± .1 | 25 ± 0 |
| | ANG | 1e-04 | .580 ± .010 | .499 ± .002 | .788 ± .002 | .794 ± .002 | 262.6 ± 2.3 | 26 ± 0 |
| | ANG | 1e-03 | .129 ± .012 | .498 ± .003 | .419 ± .015 | .427 ± .013 | 240.6 ± 4.7 | 26 ± 0 |
| | ANG | 1e-02 | .119 ± .038 | .498 ± .001 | .405 ± .048 | .411 ± .047 | 268.6 ± 1.3 | 26 ± 0 |
| | UNSIR | 1e-04 | .654 ± .013 | .498 ± .002 | .789 ± .002 | .791 ± .002 | 174.7 ± 5.8 | 27 ± 0 |
| | UNSIR | 1e-03 | .646 ± .006 | .498 ± .001 | .793 ± .001 | .796 ± .001 | 184.8 ± 4.7 | 27 ± 0 |
| | UNSIR | 1e-02 | .657 ± .005 | .499 ± .001 | .790 ± .001 | .788 ± .001 | 171.9 ± 1.5 | 26 ± 0 |
| SWM | CF−$k$ | 1e-04 | .710 ± .001 | .499 ± .001 | .792 ± .001 | .792 ± .001 | 130.2 ± 6.0 | 24 ± 0 |
| | CF−$k$ | 1e-03 | .695 ± .002 | .500 ± .000 | .794 ± .000 | .795 ± .001 | 142.6 ± 7.2 | 24 ± 0 |
| | CF−$k$ | 1e-02 | .705 ± .008 | .498 ± .000 | .793 ± .001 | .797 ± .000 | 134.0 ± 2.3 | 24 ± 0 |
| | EU−$k$ | 1e-04 | .270 ± .009 | .498 ± .001 | .788 ± .002 | .789 ± .003 | 1333.0 ± 14.0 | 24 ± 0 |
| | EU−$k$ | 1e-03 | .268 ± .005 | .499 ± .001 | .793 ± .002 | .797 ± .002 | 1359.0 ± 4.1 | 24 ± 0 |
| | EU−$k$ | 1e-02 | .267 ± .005 | .498 ± .001 | .791 ± .002 | .798 ± .002 | 1367.0 ± 39.5 | 24 ± 0 |
| | SalUn | 1e-04 | .689 ± .005 | .500 ± .001 | .788 ± .001 | .789 ± .000 | 137.7 ± 6.9 | 32 ± 0 |
| | SalUn | 1e-03 | .691 ± .015 | .500 ± .002 | .790 ± .002 | .794 ± .001 | 139.7 ± 14.6 | 31 ± 0 |
| | SalUn | 1e-02 | .698 ± .008 | .499 ± .001 | .789 ± .002 | .789 ± .001 | 131.6 ± 9.2 | 31 ± 0 |
| DIS | BT | 1e-04 | .339 ± .078 | .501 ± .003 | .593 ± .053 | .595 ± .051 | 224.5 ± 2.9 | 28 ± 0 |
| | BT | 1e-03 | .476 ± .066 | .500 ± .001 | .686 ± .045 | .692 ± .044 | 234.9 ± 3.3 | 28 ± 0 |
| | BT | 1e-02 | .256 ± .120 | .498 ± .001 | .527 ± .090 | .534 ± .093 | 224.9 ± 7.9 | 27 ± 0 |
| | SCRUB | 1e-04 | .716 ± .004 | .499 ± .002 | .792 ± .001 | .793 ± .001 | 120.9 ± 3.1 | 29 ± 0 |
| | SCRUB | 1e-03 | .715 ± .008 | .500 ± .000 | .793 ± .002 | .796 ± .001 | 121.5 ± 4.2 | 29 ± 0 |
| | SCRUB | 1e-02 | .692 ± .010 | .498 ± .002 | .787 ± .006 | .786 ± .003 | 134.8 ± .4 | 29 ± 0 |
| WI | FF | 1e-08 | .949 ± .009 | .499 ± .001 | .784 ± .001 | .791 ± .002 | 5.1 ± .7 | 18 ± 0 |
| | FF | 1e-07 | .941 ± .017 | .501 ± .001 | .785 ± .002 | .785 ± .005 | 5.6 ± 1.5 | 18 ± 0 |
| | FF | 1e-06 | .941 ± .020 | .500 ± .001 | .782 ± .009 | .786 ± .010 | 5.1 ± 1.5 | 18 ± 0 |
| | SSD | 1e-04 | .692 ± .011 | .499 ± .000 | .794 ± .002 | .795 ± .001 | 139.3 ± 4.0 | 31 ± 0 |
| | SSD | 1e-03 | .694 ± .007 | .500 ± .001 | .794 ± .002 | .795 ± .001 | 137.3 ± 4.6 | 30 ± 0 |
| | SSD | 1e-02 | .704 ± .006 | .499 ± .000 | .794 ± .002 | .795 ± .001 | 129.9 ± 1.8 | 30 ± 0 |

Table 18: Comparison of Unlearners on $MLP_{large}$ trained on Adult (mean $\pm$ std).

| Group | Method | LR | LUMA | UMIA | F1 (test) | F1 (forget) | RunTime | GPU (MB) |
|---|---|---|---|---|---|---|---|---|
| – | Orig. | - | $.631 \pm .000$ | $.498 \pm .001$ | $.793 \pm .002$ | $.798 \pm .001$ | $135.7 \pm 7.9$ | $32 \pm 0$ |
| – | Gold | - | $.636 \pm .000$ | $.499 \pm .001$ | $.791 \pm .002$ | $.792 \pm .001$ | $135.7 \pm 7.9$ | $32 \pm 0$ |
| FT | GD | 1e-04 | $.707 \pm .003$ | $.499 \pm .001$ | $.795 \pm .001$ | $.798 \pm .000$ | $87.6 \pm 3.6$ | $36 \pm 0$ |
| | GD | 1e-03 | $.710 \pm .001$ | $.498 \pm .002$ | $.791 \pm .003$ | $.795 \pm .002$ | $88.3 \pm 4.9$ | $35 \pm 0$ |
| | GD | 1e-02 | $.709 \pm .003$ | $.500 \pm .001$ | $.786 \pm .004$ | $.788 \pm .002$ | $87.2 \pm 5.3$ | $33 \pm 0$ |
| | SRL | 1e-04 | $.667 \pm .003$ | $.498 \pm .001$ | $.788 \pm .005$ | $.791 \pm .000$ | $11.0 \pm 5.8$ | $40 \pm 0$ |
| | SRL | 1e-03 | $.668 \pm .002$ | $.500 \pm .000$ | $.790 \pm .002$ | $.794 \pm .002$ | $11.2 \pm 4.7$ | $39 \pm 0$ |
| | SRL | 1e-02 | $.667 \pm .002$ | $.498 \pm .001$ | $.789 \pm .001$ | $.789 \pm .002$ | $11.6 \pm 5.6$ | $37 \pm 0$ |
| | NG | 1e-04 | $.096 \pm .085$ | $.499 \pm .002$ | $.330 \pm .136$ | $.335 \pm .135$ | $16.0 \pm .2$ | $47 \pm 0$ |
| | NG | 1e-03 | $.150 \pm .001$ | $.500 \pm .002$ | $.434 \pm .000$ | $.432 \pm .000$ | $16.2 \pm .4$ | $46 \pm 0$ |
| | NG | 1e-02 | $.108 \pm .073$ | $.499 \pm .001$ | $.352 \pm .141$ | $.352 \pm .137$ | $16.4 \pm .3$ | $44 \pm 0$ |
| | ANG | 1e-04 | $.591 \pm .006$ | $.499 \pm .000$ | $.767 \pm .004$ | $.773 \pm .003$ | $140.7 \pm 1.5$ | $50 \pm 0$ |
| | ANG | 1e-03 | $.117 \pm .014$ | $.499 \pm .001$ | $.403 \pm .018$ | $.409 \pm .017$ | $144.6 \pm 2.3$ | $49 \pm 0$ |
| | ANG | 1e-02 | $.062 \pm .066$ | $.501 \pm .004$ | $.271 \pm .141$ | $.273 \pm .137$ | $148.6 \pm 1.1$ | $48 \pm 0$ |
| | UNSIR | 1e-04 | $.666 \pm .010$ | $.499 \pm .001$ | $.791 \pm .002$ | $.797 \pm .001$ | $105.6 \pm .0$ | $54 \pm 0$ |
| | UNSIR | 1e-03 | $.668 \pm .008$ | $.498 \pm .001$ | $.792 \pm .003$ | $.795 \pm .002$ | $105.2 \pm .6$ | $53 \pm 0$ |
| | UNSIR | 1e-02 | $.668 \pm .008$ | $.499 \pm .001$ | $.787 \pm .004$ | $.790 \pm .002$ | $105.2 \pm .8$ | $51 \pm 0$ |
| SWM | CF$-k$ | 1e-04 | $.711 \pm .007$ | $.499 \pm .000$ | $.791 \pm .003$ | $.797 \pm .001$ | $85.5 \pm 1.1$ | $41 \pm 0$ |
| | CF$-k$ | 1e-03 | $.710 \pm .007$ | $.500 \pm .001$ | $.793 \pm .003$ | $.797 \pm .002$ | $85.2 \pm .9$ | $41 \pm 0$ |
| | CF$-k$ | 1e-02 | $.706 \pm .002$ | $.498 \pm .001$ | $.794 \pm .003$ | $.797 \pm .000$ | $87.7 \pm 5.3$ | $41 \pm 0$ |
| | EU$-k$ | 1e-04 | $.240 \pm .012$ | $.499 \pm .001$ | $.789 \pm .003$ | $.793 \pm .004$ | $850.2 \pm 6.4$ | $44 \pm 0$ |
| | EU$-k$ | 1e-03 | $.242 \pm .010$ | $.498 \pm .001$ | $.793 \pm .002$ | $.796 \pm .002$ | $848.4 \pm 15.6$ | $43 \pm 0$ |
| | EU$-k$ | 1e-02 | $.241 \pm .009$ | $.500 \pm .000$ | $.794 \pm .004$ | $.796 \pm .001$ | $856.0 \pm 16.2$ | $42 \pm 0$ |
| | SalUn | 1e-04 | $.676 \pm .014$ | $.500 \pm .001$ | $.786 \pm .003$ | $.790 \pm .001$ | $100.8 \pm 11.9$ | $57 \pm 27$ |
| | SalUn | 1e-03 | $.675 \pm .018$ | $.499 \pm .000$ | $.791 \pm .001$ | $.796 \pm .002$ | $103.1 \pm 14.3$ | $55 \pm 27$ |
| | SalUn | 1e-02 | $.676 \pm .016$ | $.498 \pm .001$ | $.792 \pm .001$ | $.791 \pm .003$ | $103.7 \pm 14.3$ | $53 \pm 26$ |
| DIS | BT | 1e-04 | $.354 \pm .260$ | $.499 \pm .003$ | $.576 \pm .247$ | $.578 \pm .238$ | $155.1 \pm 1.6$ | $58 \pm 0$ |
| | BT | 1e-03 | $.432 \pm .154$ | $.504 \pm .005$ | $.660 \pm .106$ | $.669 \pm .112$ | $156.1 \pm .9$ | $57 \pm 0$ |
| | BT | 1e-02 | $.427 \pm .175$ | $.502 \pm .004$ | $.661 \pm .127$ | $.663 \pm .134$ | $155.5 \pm 1.8$ | $56 \pm 0$ |
| | SCRUB | 1e-04 | $.708 \pm .009$ | $.498 \pm .001$ | $.795 \pm .001$ | $.799 \pm .000$ | $80.8 \pm .2$ | $61 \pm 0$ |
| | SCRUB | 1e-03 | $.711 \pm .007$ | $.498 \pm .002$ | $.789 \pm .003$ | $.791 \pm .004$ | $81.4 \pm 1.0$ | $60 \pm 0$ |
| | SCRUB | 1e-02 | $.708 \pm .012$ | $.499 \pm .003$ | $.790 \pm .005$ | $.791 \pm .007$ | $82.1 \pm .4$ | $60 \pm 0$ |
| WI | FF | 1e-08 | $.897 \pm .046$ | $.501 \pm .002$ | $.773 \pm .018$ | $.770 \pm .021$ | $8.3 \pm .8$ | $21 \pm 0$ |
| | FF | 1e-07 | $.936 \pm .013$ | $.500 \pm .001$ | $.786 \pm .007$ | $.785 \pm .006$ | $8.0 \pm .8$ | $20 \pm 0$ |
| | FF | 1e-06 | $.897 \pm .048$ | $.500 \pm .002$ | $.768 \pm .017$ | $.770 \pm .020$ | $7.2 \pm .4$ | $19 \pm 0$ |
| | SSD | 1e-04 | $.676 \pm .023$ | $.499 \pm .000$ | $.793 \pm .002$ | $.798 \pm .001$ | $103.7 \pm 19.6$ | $52 \pm 26$ |
| | SSD | 1e-03 | $.675 \pm .023$ | $.499 \pm .001$ | $.793 \pm .002$ | $.798 \pm .001$ | $104.4 \pm 2.0$ | $51 \pm 26$ |
| | SSD | 1e-02 | $.697 \pm .016$ | $.499 \pm .001$ | $.793 \pm .002$ | $.798 \pm .001$ | $91.5 \pm .9$ | $50 \pm 26$ |

Table 19: Comparison of Unlearners on $MLP_{small}$ trained on Spotify (mean $\pm$ std).

| Group | Method | LR | LUMA | UMIA | F1 (test) | F1 (forget) | RunTime | GPU (MB) |
|---|---|---|---|---|---|---|---|---|
| – | Orig. | - | $0.613 \pm 0.002$ | $0.499 \pm 0.004$ | $0.612 \pm 0.002$ | $0.620 \pm 0.004$ | $189.8 \pm 4.2$ | $19 \pm 0$ |
| – | Gold | - | $0.636 \pm 0.000$ | $0.499 \pm 0.006$ | $0.606 \pm 0.002$ | $0.588 \pm 0.001$ | $189.8 \pm 4.2$ | $19 \pm 0$ |
| FT | GD | 1e-04 | $0.890 \pm 0.009$ | $0.500 \pm 0.004$ | $0.615 \pm 0.003$ | $0.624 \pm 0.007$ | $10.0 \pm 0.5$ | $19 \pm 0$ |
| | GD | 1e-03 | $0.897 \pm 0.005$ | $0.498 \pm 0.003$ | $0.614 \pm 0.005$ | $0.615 \pm 0.004$ | $10.1 \pm 0.5$ | $19 \pm 0$ |
| | GD | 1e-02 | $0.939 \pm 0.002$ | $0.497 \pm 0.002$ | $0.609 \pm 0.003$ | $0.602 \pm 0.001$ | $4.6 \pm 0.2$ | $18 \pm 0$ |
| | SRL | 1e-04 | $0.879 \pm 0.007$ | $0.497 \pm 0.004$ | $0.615 \pm 0.002$ | $0.625 \pm 0.006$ | $12.3 \pm 1.0$ | $20 \pm 0$ |
| | SRL | 1e-03 | $0.890 \pm 0.004$ | $0.499 \pm 0.001$ | $0.612 \pm 0.002$ | $0.617 \pm 0.003$ | $12.7 \pm 0.4$ | $19 \pm 0$ |
| | SRL | 1e-02 | $0.936 \pm 0.003$ | $0.496 \pm 0.001$ | $0.599 \pm 0.005$ | $0.590 \pm 0.006$ | $6.6 \pm 0.3$ | $18 \pm 0$ |
| | NG | 1e-04 | $0.918 \pm 0.009$ | $0.498 \pm 0.002$ | $0.613 \pm 0.004$ | $0.618 \pm 0.007$ | $2.4 \pm 0.4$ | $20 \pm 0$ |
| | NG | 1e-03 | $0.836 \pm 0.038$ | $0.496 \pm 0.006$ | $0.551 \pm 0.018$ | $0.546 \pm 0.006$ | $2.4 \pm 0.3$ | $20 \pm 0$ |
| | NG | 1e-02 | $0.038 \pm 0.007$ | $0.501 \pm 0.011$ | $0.053 \pm 0.017$ | $0.052 \pm 0.024$ | $1.3 \pm 0.1$ | $18 \pm 0$ |
| | ANG | 1e-04 | $0.869 \pm 0.008$ | $0.497 \pm 0.001$ | $0.614 \pm 0.006$ | $0.610 \pm 0.003$ | $20.0 \pm 0.6$ | $21 \pm 0$ |
| | ANG | 1e-03 | $0.659 \pm 0.012$ | $0.502 \pm 0.011$ | $0.505 \pm 0.004$ | $0.492 \pm 0.008$ | $19.3 \pm 1.6$ | $21 \pm 0$ |
| | ANG | 1e-02 | $0.078 \pm 0.014$ | $0.529 \pm 0.007$ | $0.161 \pm 0.029$ | $0.133 \pm 0.021$ | $10.2 \pm 0.3$ | $18 \pm 0$ |
| | UNSIR | 1e-04 | $0.862 \pm 0.011$ | $0.499 \pm 0.004$ | $0.614 \pm 0.006$ | $0.626 \pm 0.005$ | $15.7 \pm 0.8$ | $21 \pm 0$ |
| | UNSIR | 1e-03 | $0.884 \pm 0.005$ | $0.497 \pm 0.002$ | $0.609 \pm 0.005$ | $0.615 \pm 0.005$ | $15.2 \pm 0.9$ | $21 \pm 0$ |
| | UNSIR | 1e-02 | $0.919 \pm 0.010$ | $0.495 \pm 0.002$ | $0.607 \pm 0.006$ | $0.602 \pm 0.004$ | $8.0 \pm 0.3$ | $19 \pm 0$ |
| SWM | $CF-k$ | 1e-04 | $0.881 \pm 0.004$ | $0.498 \pm 0.001$ | $0.615 \pm 0.001$ | $0.626 \pm 0.006$ | $10.3 \pm 0.4$ | $20 \pm 0$ |
| | $CF-k$ | 1e-03 | $0.888 \pm 0.008$ | $0.499 \pm 0.000$ | $0.614 \pm 0.003$ | $0.621 \pm 0.007$ | $10.3 \pm 0.5$ | $20 \pm 0$ |
| | $CF-k$ | 1e-02 | $0.931 \pm 0.010$ | $0.501 \pm 0.002$ | $0.603 \pm 0.003$ | $0.608 \pm 0.007$ | $5.2 \pm 0.3$ | $18 \pm 0$ |
| | $EU-k$ | 1e-04 | $0.517 \pm 0.021$ | $0.496 \pm 0.004$ | $0.480 \pm 0.009$ | $0.458 \pm 0.013$ | $93.8 \pm 4.4$ | $20 \pm 0$ |
| | $EU-k$ | 1e-03 | $0.734 \pm 0.012$ | $0.497 \pm 0.001$ | $0.598 \pm 0.002$ | $0.587 \pm 0.005$ | $97.5 \pm 6.5$ | $20 \pm 0$ |
| | $EU-k$ | 1e-02 | $0.796 \pm 0.002$ | $0.497 \pm 0.003$ | $0.613 \pm 0.005$ | $0.614 \pm 0.001$ | $51.7 \pm 2.2$ | $18 \pm 0$ |
| | SalUn | 1e-04 | $0.874 \pm 0.004$ | $0.501 \pm 0.002$ | $0.615 \pm 0.002$ | $0.624 \pm 0.008$ | $13.7 \pm 1.0$ | $23 \pm 0$ |
| | SalUn | 1e-03 | $0.883 \pm 0.014$ | $0.498 \pm 0.003$ | $0.612 \pm 0.003$ | $0.619 \pm 0.008$ | $14.3 \pm 1.0$ | $23 \pm 0$ |
| | SalUn | 1e-02 | $0.931 \pm 0.008$ | $0.496 \pm 0.001$ | $0.601 \pm 0.004$ | $0.595 \pm 0.004$ | $8.0 \pm 0.2$ | $19 \pm 0$ |
| DIS | BT | 1e-04 | $0.879 \pm 0.012$ | $0.501 \pm 0.005$ | $0.595 \pm 0.011$ | $0.597 \pm 0.006$ | $20.9 \pm 1.3$ | $22 \pm 0$ |
| | BT | 1e-03 | $0.455 \pm 0.110$ | $0.506 \pm 0.007$ | $0.403 \pm 0.046$ | $0.429 \pm 0.050$ | $23.7 \pm 1.9$ | $21 \pm 0$ |
| | BT | 1e-02 | $0.545 \pm 0.103$ | $0.505 \pm 0.002$ | $0.439 \pm 0.032$ | $0.456 \pm 0.045$ | $11.8 \pm 0.8$ | $19 \pm 0$ |
| | SCRUB | 1e-04 | $0.874 \pm 0.009$ | $0.496 \pm 0.002$ | $0.615 \pm 0.004$ | $0.625 \pm 0.006$ | $11.8 \pm 0.7$ | $22 \pm 0$ |
| | SCRUB | 1e-03 | $0.882 \pm 0.010$ | $0.498 \pm 0.001$ | $0.617 \pm 0.003$ | $0.617 \pm 0.006$ | $12.0 \pm 0.9$ | $22 \pm 0$ |
| | SCRUB | 1e-02 | $0.784 \pm 0.059$ | $0.499 \pm 0.001$ | $0.533 \pm 0.017$ | $0.534 \pm 0.027$ | $6.5 \pm 0.3$ | $19 \pm 0$ |
| WI | FF | 1e-08 | $0.890 \pm 0.002$ | $0.497 \pm 0.001$ | $0.593 \pm 0.013$ | $0.603 \pm 0.009$ | $15.1 \pm 0.8$ | $18 \pm 0$ |
| | FF | 1e-07 | $0.898 \pm 0.007$ | $0.497 \pm 0.006$ | $0.594 \pm 0.009$ | $0.599 \pm 0.008$ | $15.5 \pm 0.2$ | $18 \pm 0$ |
| | FF | 1e-06 | $0.887 \pm 0.015$ | $0.497 \pm 0.006$ | $0.594 \pm 0.002$ | $0.600 \pm 0.011$ | $15.8 \pm 0.1$ | $18 \pm 0$ |
| | SSD | 1e-04 | $0.877 \pm 0.004$ | $0.501 \pm 0.004$ | $0.612 \pm 0.002$ | $0.620 \pm 0.004$ | $14.5 \pm 0.8$ | $22 \pm 0$ |
| | SSD | 1e-03 | $0.883 \pm 0.008$ | $0.500 \pm 0.006$ | $0.612 \pm 0.002$ | $0.620 \pm 0.004$ | $13.7 \pm 0.8$ | $22 \pm 0$ |
| | SSD | 1e-02 | $0.901 \pm 0.004$ | $0.501 \pm 0.003$ | $0.612 \pm 0.002$ | $0.620 \pm 0.004$ | $8.0 \pm 0.6$ | $19 \pm 0$ |

Table 20: Comparison of Unlearners on $MLP_{large}$ trained on Spotify (mean ± std).

| Group | Method | LR | LUMA | UMIA | F1 (test) | F1 (forget) | RunTime | GPU (MB) |
|---|---|---|---|---|---|---|---|---|
| − | Orig. | - | $0.573 \pm 0.007$ | $0.528 \pm 0.005$ | $0.635 \pm 0.009$ | $0.689 \pm 0.008$ | $119.6 \pm 1.7$ | $30 \pm 0$ |
| − | Gold | - | $0.636 \pm 0.000$ | $0.497 \pm 0.003$ | $0.629 \pm 0.002$ | $0.622 \pm 0.004$ | $119.6 \pm 1.7$ | $30 \pm 0$ |
| FT | GD | 1e-04 | $0.810 \pm 0.003$ | $0.526 \pm 0.003$ | $0.642 \pm 0.007$ | $0.695 \pm 0.003$ | $6.0 \pm 0.1$ | $33 \pm 0$ |
| | GD | 1e-03 | $0.850 \pm 0.022$ | $0.516 \pm 0.009$ | $0.638 \pm 0.008$ | $0.676 \pm 0.006$ | $6.0 \pm 0.1$ | $31 \pm 0$ |
| | GD | 1e-02 | $0.915 \pm 0.018$ | $0.498 \pm 0.004$ | $0.604 \pm 0.007$ | $0.614 \pm 0.012$ | $4.7 \pm 0.1$ | $22 \pm 0$ |
| | SRL | 1e-04 | $0.796 \pm 0.006$ | $0.530 \pm 0.000$ | $0.643 \pm 0.004$ | $0.696 \pm 0.002$ | $7.6 \pm 0.1$ | $36 \pm 0$ |
| | SRL | 1e-03 | $0.842 \pm 0.012$ | $0.521 \pm 0.002$ | $0.634 \pm 0.007$ | $0.675 \pm 0.008$ | $7.8 \pm 0.1$ | $35 \pm 0$ |
| | SRL | 1e-02 | $0.764 \pm 0.002$ | $0.494 \pm 0.004$ | $0.555 \pm 0.001$ | $0.554 \pm 0.004$ | $6.8 \pm 0.0$ | $23 \pm 0$ |
| | NG | 1e-04 | $0.875 \pm 0.006$ | $0.522 \pm 0.008$ | $0.604 \pm 0.016$ | $0.644 \pm 0.016$ | $1.5 \pm 0.0$ | $42 \pm 0$ |
| | NG | 1e-03 | $0.023 \pm 0.000$ | $0.504 \pm 0.011$ | $0.020 \pm 0.000$ | $0.021 \pm 0.000$ | $1.5 \pm 0.0$ | $40 \pm 0$ |
| | NG | 1e-02 | $0.023 \pm 0.000$ | $0.502 \pm 0.002$ | $0.020 \pm 0.000$ | $0.021 \pm 0.001$ | $1.3 \pm 0.0$ | $25 \pm 0$ |
| | ANG | 1e-04 | $0.875 \pm 0.002$ | $0.503 \pm 0.003$ | $0.608 \pm 0.009$ | $0.635 \pm 0.013$ | $11.9 \pm 0.1$ | $45 \pm 0$ |
| | ANG | 1e-03 | $0.085 \pm 0.021$ | $0.534 \pm 0.003$ | $0.189 \pm 0.031$ | $0.183 \pm 0.030$ | $11.8 \pm 0.1$ | $44 \pm 0$ |
| | ANG | 1e-02 | $0.024 \pm 0.001$ | $0.521 \pm 0.003$ | $0.024 \pm 0.002$ | $0.026 \pm 0.002$ | $10.7 \pm 0.3$ | $26 \pm 0$ |
| | UNSIR | 1e-04 | $0.792 \pm 0.004$ | $0.528 \pm 0.001$ | $0.643 \pm 0.005$ | $0.694 \pm 0.003$ | $9.3 \pm 0.0$ | $48 \pm 0$ |
| | UNSIR | 1e-03 | $0.824 \pm 0.011$ | $0.519 \pm 0.006$ | $0.642 \pm 0.008$ | $0.677 \pm 0.009$ | $9.4 \pm 0.1$ | $47 \pm 0$ |
| | UNSIR | 1e-02 | $0.885 \pm 0.012$ | $0.499 \pm 0.001$ | $0.603 \pm 0.007$ | $0.602 \pm 0.010$ | $8.5 \pm 0.2$ | $27 \pm 0$ |
| SWM | CF$-k$ | 1e-04 | $0.810 \pm 0.010$ | $0.529 \pm 0.003$ | $0.641 \pm 0.005$ | $0.691 \pm 0.007$ | $5.9 \pm 0.1$ | $37 \pm 0$ |
| | CF$-k$ | 1e-03 | $0.809 \pm 0.011$ | $0.523 \pm 0.009$ | $0.644 \pm 0.004$ | $0.695 \pm 0.001$ | $6.0 \pm 0.1$ | $38 \pm 0$ |
| | CF$-k$ | 1e-02 | $0.829 \pm 0.011$ | $0.519 \pm 0.007$ | $0.642 \pm 0.007$ | $0.690 \pm 0.007$ | $5.3 \pm 0.1$ | $23 \pm 0$ |
| | EU$-k$ | 1e-04 | $0.687 \pm 0.009$ | $0.508 \pm 0.008$ | $0.582 \pm 0.009$ | $0.595 \pm 0.002$ | $59.6 \pm 0.5$ | $39 \pm 0$ |
| | EU$-k$ | 1e-03 | $0.692 \pm 0.000$ | $0.514 \pm 0.007$ | $0.642 \pm 0.007$ | $0.673 \pm 0.005$ | $59.6 \pm 0.3$ | $39 \pm 0$ |
| | EU$-k$ | 1e-02 | $0.687 \pm 0.009$ | $0.521 \pm 0.002$ | $0.646 \pm 0.006$ | $0.693 \pm 0.008$ | $52.8 \pm 0.2$ | $24 \pm 0$ |
| | SalUn | 1e-04 | $0.786 \pm 0.009$ | $0.530 \pm 0.003$ | $0.645 \pm 0.006$ | $0.697 \pm 0.002$ | $8.0 \pm 0.6$ | $63 \pm 0$ |
| | SalUn | 1e-03 | $0.817 \pm 0.021$ | $0.526 \pm 0.006$ | $0.635 \pm 0.009$ | $0.683 \pm 0.009$ | $8.3 \pm 0.6$ | $60 \pm 0$ |
| | SalUn | 1e-02 | $0.779 \pm 0.026$ | $0.501 \pm 0.004$ | $0.564 \pm 0.006$ | $0.562 \pm 0.014$ | $8.4 \pm 0.3$ | $31 \pm 0$ |
| DIS | BT | 1e-04 | $0.785 \pm 0.029$ | $0.531 \pm 0.003$ | $0.583 \pm 0.027$ | $0.634 \pm 0.032$ | $13.2 \pm 1.0$ | $51 \pm 0$ |
| | BT | 1e-03 | $0.641 \pm 0.114$ | $0.531 \pm 0.002$ | $0.501 \pm 0.052$ | $0.554 \pm 0.050$ | $13.6 \pm 0.8$ | $50 \pm 0$ |
| | BT | 1e-02 | $0.667 \pm 0.060$ | $0.521 \pm 0.006$ | $0.520 \pm 0.019$ | $0.543 \pm 0.024$ | $12.8 \pm 0.7$ | $28 \pm 0$ |
| | SCRUB | 1e-04 | $0.796 \pm 0.013$ | $0.524 \pm 0.006$ | $0.647 \pm 0.003$ | $0.696 \pm 0.006$ | $7.6 \pm 0.1$ | $54 \pm 0$ |
| | SCRUB | 1e-03 | $0.841 \pm 0.004$ | $0.515 \pm 0.007$ | $0.611 \pm 0.013$ | $0.667 \pm 0.014$ | $7.6 \pm 0.1$ | $54 \pm 0$ |
| | SCRUB | 1e-02 | $0.325 \pm 0.072$ | $0.498 \pm 0.000$ | $0.382 \pm 0.035$ | $0.377 \pm 0.041$ | $6.8 \pm 0.1$ | $29 \pm 0$ |
| WI | FF | 1e-08 | $0.808 \pm 0.020$ | $0.523 \pm 0.007$ | $0.599 \pm 0.025$ | $0.646 \pm 0.016$ | $19.3 \pm 0.6$ | $21 \pm 0$ |
| | FF | 1e-07 | $0.812 \pm 0.017$ | $0.517 \pm 0.008$ | $0.600 \pm 0.028$ | $0.640 \pm 0.033$ | $19.4 \pm 0.2$ | $20 \pm 0$ |
| | FF | 1e-06 | $0.769 \pm 0.107$ | $0.513 \pm 0.009$ | $0.572 \pm 0.051$ | $0.605 \pm 0.063$ | $19.3 \pm 0.2$ | $19 \pm 0$ |
| | SSD | 1e-04 | $0.807 \pm 0.018$ | $0.528 \pm 0.003$ | $0.635 \pm 0.009$ | $0.689 \pm 0.008$ | $8.7 \pm 0.2$ | $59 \pm 0$ |
| | SSD | 1e-03 | $0.806 \pm 0.013$ | $0.528 \pm 0.003$ | $0.635 \pm 0.009$ | $0.689 \pm 0.008$ | $8.9 \pm 0.1$ | $58 \pm 0$ |
| | SSD | 1e-02 | $0.827 \pm 0.017$ | $0.518 \pm 0.007$ | $0.635 \pm 0.009$ | $0.689 \pm 0.008$ | $8.1 \pm 0.1$ | $30 \pm 0$ |

Table 21: Comparison of Unlearners on $GCN_{small}$ trained on BACE (mean $\pm$ std).

| Group | Method | LR | LUMA | UMIA | F1 (test) | F1 (forget) | RunTime | GPU (MB) |
|---|---|---|---|---|---|---|---|---|
| – | Orig. | - | $0.594 \pm 0.046$ | $0.535 \pm 0.019$ | $0.572 \pm 0.009$ | $0.662 \pm 0.011$ | $70.2 \pm 0.3$ | $18 \pm 0$ |
| – | Gold | - | $0.636 \pm 0.000$ | $0.521 \pm 0.015$ | $0.540 \pm 0.019$ | $0.641 \pm 0.019$ | $70.2 \pm 0.3$ | $18 \pm 0$ |
| FT | GD | 1e-04 | $0.881 \pm 0.077$ | $0.520 \pm 0.013$ | $0.570 \pm 0.007$ | $0.663 \pm 0.012$ | $1.3 \pm 0.0$ | $18 \pm 0$ |
| | GD | 1e-03 | $0.897 \pm 0.095$ | $0.528 \pm 0.009$ | $0.566 \pm 0.009$ | $0.661 \pm 0.020$ | $1.3 \pm 0.0$ | $18 \pm 0$ |
| | GD | 1e-02 | $0.894 \pm 0.013$ | $0.505 \pm 0.022$ | $0.552 \pm 0.024$ | $0.644 \pm 0.015$ | $1.3 \pm 0.0$ | $18 \pm 0$ |
| | SRL | 1e-04 | $0.876 \pm 0.073$ | $0.536 \pm 0.004$ | $0.572 \pm 0.004$ | $0.663 \pm 0.010$ | $1.8 \pm 0.0$ | $18 \pm 0$ |
| | SRL | 1e-03 | $0.890 \pm 0.006$ | $0.505 \pm 0.012$ | $0.536 \pm 0.005$ | $0.646 \pm 0.019$ | $1.8 \pm 0.0$ | $18 \pm 0$ |
| | SRL | 1e-02 | $0.336 \pm 0.039$ | $0.499 \pm 0.003$ | $0.355 \pm 0.000$ | $0.348 \pm 0.000$ | $1.7 \pm 0.0$ | $18 \pm 0$ |
| | NG | 1e-04 | $0.596 \pm 0.054$ | $0.494 \pm 0.001$ | $0.414 \pm 0.004$ | $0.508 \pm 0.019$ | $0.4 \pm 0.0$ | $18 \pm 0$ |
| | NG | 1e-03 | $0.336 \pm 0.041$ | $0.499 \pm 0.014$ | $0.355 \pm 0.000$ | $0.348 \pm 0.000$ | $0.4 \pm 0.0$ | $18 \pm 0$ |
| | NG | 1e-02 | $0.338 \pm 0.040$ | $0.505 \pm 0.011$ | $0.355 \pm 0.000$ | $0.348 \pm 0.000$ | $0.4 \pm 0.0$ | $18 \pm 0$ |
| | ANG | 1e-04 | $0.892 \pm 0.005$ | $0.512 \pm 0.015$ | $0.537 \pm 0.008$ | $0.634 \pm 0.018$ | $1.0 \pm 0.0$ | $18 \pm 0$ |
| | ANG | 1e-03 | $0.395 \pm 0.027$ | $0.495 \pm 0.011$ | $0.379 \pm 0.008$ | $0.380 \pm 0.011$ | $1.0 \pm 0.0$ | $18 \pm 0$ |
| | ANG | 1e-02 | $0.337 \pm 0.039$ | $0.507 \pm 0.012$ | $0.355 \pm 0.000$ | $0.348 \pm 0.000$ | $1.0 \pm 0.0$ | $18 \pm 0$ |
| | UNSIR | 1e-04 | $0.885 \pm 0.076$ | $0.519 \pm 0.006$ | $0.570 \pm 0.007$ | $0.663 \pm 0.012$ | $1.6 \pm 0.0$ | $18 \pm 0$ |
| | UNSIR | 1e-03 | $0.886 \pm 0.088$ | $0.504 \pm 0.017$ | $0.566 \pm 0.009$ | $0.661 \pm 0.020$ | $1.6 \pm 0.0$ | $18 \pm 0$ |
| | UNSIR | 1e-02 | $0.900 \pm 0.010$ | $0.525 \pm 0.012$ | $0.552 \pm 0.024$ | $0.644 \pm 0.015$ | $1.6 \pm 0.0$ | $18 \pm 0$ |
| SWM | CF$-k$ | 1e-04 | $0.880 \pm 0.067$ | $0.514 \pm 0.011$ | $0.570 \pm 0.007$ | $0.663 \pm 0.012$ | $1.4 \pm 0.0$ | $18 \pm 0$ |
| | CF$-k$ | 1e-03 | $0.883 \pm 0.067$ | $0.508 \pm 0.010$ | $0.566 \pm 0.009$ | $0.661 \pm 0.020$ | $1.4 \pm 0.0$ | $18 \pm 0$ |
| | CF$-k$ | 1e-02 | $0.911 \pm 0.014$ | $0.513 \pm 0.008$ | $0.552 \pm 0.024$ | $0.644 \pm 0.015$ | $1.4 \pm 0.0$ | $18 \pm 0$ |
| | EU$-k$ | 1e-04 | $0.577 \pm 0.040$ | $0.515 \pm 0.038$ | $0.573 \pm 0.011$ | $0.668 \pm 0.012$ | $69.7 \pm 0.4$ | $18 \pm 0$ |
| | EU$-k$ | 1e-03 | $0.585 \pm 0.041$ | $0.519 \pm 0.020$ | $0.577 \pm 0.007$ | $0.671 \pm 0.017$ | $69.4 \pm 1.7$ | $18 \pm 0$ |
| | EU$-k$ | 1e-02 | $0.542 \pm 0.037$ | $0.493 \pm 0.004$ | $0.624 \pm 0.018$ | $0.681 \pm 0.008$ | $70.1 \pm 0.3$ | $18 \pm 0$ |
| | SalUn | 1e-04 | $0.869 \pm 0.061$ | $0.521 \pm 0.014$ | $0.572 \pm 0.004$ | $0.663 \pm 0.010$ | $2.1 \pm 0.0$ | $19 \pm 0$ |
| | SalUn | 1e-03 | $0.900 \pm 0.011$ | $0.510 \pm 0.010$ | $0.539 \pm 0.005$ | $0.643 \pm 0.017$ | $2.1 \pm 0.0$ | $19 \pm 0$ |
| | SalUn | 1e-02 | $0.335 \pm 0.038$ | $0.497 \pm 0.008$ | $0.355 \pm 0.000$ | $0.348 \pm 0.000$ | $2.1 \pm 0.0$ | $19 \pm 0$ |
| DIS | BT | 1e-04 | $0.792 \pm 0.077$ | $0.559 \pm 0.007$ | $0.588 \pm 0.008$ | $0.689 \pm 0.017$ | $3.2 \pm 0.0$ | $19 \pm 0$ |
| | BT | 1e-03 | $0.836 \pm 0.117$ | $0.535 \pm 0.015$ | $0.589 \pm 0.022$ | $0.669 \pm 0.021$ | $3.2 \pm 0.0$ | $19 \pm 0$ |
| | BT | 1e-02 | $0.777 \pm 0.010$ | $0.499 \pm 0.016$ | $0.503 \pm 0.010$ | $0.556 \pm 0.021$ | $3.2 \pm 0.0$ | $19 \pm 0$ |
| | SCRUB | 1e-04 | $0.889 \pm 0.077$ | $0.522 \pm 0.009$ | $0.570 \pm 0.007$ | $0.663 \pm 0.012$ | $2.3 \pm 0.0$ | $19 \pm 0$ |
| | SCRUB | 1e-03 | $0.875 \pm 0.093$ | $0.526 \pm 0.017$ | $0.566 \pm 0.009$ | $0.661 \pm 0.020$ | $2.4 \pm 0.1$ | $19 \pm 0$ |
| | SCRUB | 1e-02 | $0.893 \pm 0.028$ | $0.505 \pm 0.023$ | $0.552 \pm 0.024$ | $0.644 \pm 0.015$ | $2.3 \pm 0.0$ | $19 \pm 0$ |
| WI | FF | 1e-08 | $0.547 \pm 0.192$ | $0.509 \pm 0.027$ | $0.417 \pm 0.048$ | $0.483 \pm 0.085$ | $10.1 \pm 0.1$ | $19 \pm 0$ |
| | FF | 1e-07 | $0.609 \pm 0.267$ | $0.547 \pm 0.039$ | $0.498 \pm 0.117$ | $0.569 \pm 0.192$ | $10.0 \pm 0.0$ | $19 \pm 0$ |
| | FF | 1e-06 | $0.673 \pm 0.197$ | $0.541 \pm 0.040$ | $0.523 \pm 0.125$ | $0.573 \pm 0.155$ | $10.0 \pm 0.1$ | $19 \pm 0$ |
| | SSD | 1e-04 | $0.888 \pm 0.062$ | $0.526 \pm 0.014$ | $0.572 \pm 0.009$ | $0.662 \pm 0.011$ | $2.0 \pm 0.0$ | $19 \pm 0$ |
| | SSD | 1e-03 | $0.876 \pm 0.064$ | $0.527 \pm 0.009$ | $0.572 \pm 0.009$ | $0.662 \pm 0.011$ | $1.9 \pm 0.0$ | $19 \pm 0$ |
| | SSD | 1e-02 | $0.865 \pm 0.094$ | $0.527 \pm 0.023$ | $0.572 \pm 0.009$ | $0.662 \pm 0.011$ | $1.9 \pm 0.0$ | $19 \pm 0$ |

Table 22: Comparison of Unlearners on $GCN_{large}$ trained on BACE (mean ± std).

| Group | Method | LR | LUMA | UMIA | F1 (test) | F1 (forget) | RunTime | GPU (MB) |
|---|---|---|---|---|---|---|---|---|
| – | Orig. | - | $0.564 \pm 0.048$ | $0.525 \pm 0.025$ | $0.630 \pm 0.004$ | $0.711 \pm 0.002$ | $57.3 \pm 27.8$ | $18 \pm 0$ |
| – | Gold | - | $0.636 \pm 0.000$ | $0.528 \pm 0.009$ | $0.559 \pm 0.044$ | $0.679 \pm 0.026$ | $57.3 \pm 27.8$ | $18 \pm 0$ |
| FT | GD | 1e-04 | $0.851 \pm 0.102$ | $0.519 \pm 0.017$ | $0.612 \pm 0.021$ | $0.698 \pm 0.003$ | $3.1 \pm 4.0$ | $19 \pm 0$ |
| | GD | 1e-03 | $0.837 \pm 0.083$ | $0.534 \pm 0.008$ | $0.617 \pm 0.009$ | $0.707 \pm 0.005$ | $3.0 \pm 4.0$ | $19 \pm 0$ |
| | GD | 1e-02 | $0.444 \pm 0.302$ | $0.499 \pm 0.005$ | $0.435 \pm 0.123$ | $0.468 \pm 0.180$ | $1.1 \pm 0.6$ | $18 \pm 0$ |
| | SRL | 1e-04 | $0.832 \pm 0.089$ | $0.536 \pm 0.017$ | $0.609 \pm 0.015$ | $0.701 \pm 0.009$ | $4.0 \pm 5.1$ | $19 \pm 0$ |
| | SRL | 1e-03 | $0.826 \pm 0.118$ | $0.504 \pm 0.031$ | $0.603 \pm 0.023$ | $0.694 \pm 0.011$ | $4.1 \pm 5.2$ | $19 \pm 0$ |
| | SRL | 1e-02 | $0.478 \pm 0.402$ | $0.510 \pm 0.022$ | $0.411 \pm 0.096$ | $0.447 \pm 0.171$ | $3.9 \pm 5.0$ | $19 \pm 0$ |
| | NG | 1e-04 | $0.368 \pm 0.033$ | $0.488 \pm 0.011$ | $0.385 \pm 0.026$ | $0.410 \pm 0.057$ | $0.2 \pm 0.0$ | $19 \pm 0$ |
| | NG | 1e-03 | $0.283 \pm 0.060$ | $0.499 \pm 0.008$ | $0.355 \pm 0.000$ | $0.348 \pm 0.000$ | $0.2 \pm 0.0$ | $19 \pm 0$ |
| | NG | 1e-02 | $0.265 \pm 0.081$ | $0.506 \pm 0.009$ | $0.340 \pm 0.026$ | $0.338 \pm 0.017$ | $0.2 \pm 0.0$ | $19 \pm 0$ |
| | ANG | 1e-04 | $0.833 \pm 0.099$ | $0.505 \pm 0.020$ | $0.620 \pm 0.009$ | $0.702 \pm 0.003$ | $0.7 \pm 0.0$ | $20 \pm 0$ |
| | ANG | 1e-03 | $0.869 \pm 0.031$ | $0.496 \pm 0.017$ | $0.534 \pm 0.028$ | $0.658 \pm 0.015$ | $0.7 \pm 0.0$ | $19 \pm 0$ |
| | ANG | 1e-02 | $0.284 \pm 0.063$ | $0.503 \pm 0.007$ | $0.355 \pm 0.000$ | $0.348 \pm 0.000$ | $0.7 \pm 0.0$ | $19 \pm 0$ |
| | UNSIR | 1e-04 | $0.843 \pm 0.104$ | $0.513 \pm 0.029$ | $0.612 \pm 0.021$ | $0.698 \pm 0.003$ | $1.0 \pm 0.0$ | $20 \pm 0$ |
| | UNSIR | 1e-03 | $0.850 \pm 0.097$ | $0.528 \pm 0.006$ | $0.617 \pm 0.009$ | $0.707 \pm 0.005$ | $1.0 \pm 0.0$ | $20 \pm 0$ |
| | UNSIR | 1e-02 | $0.458 \pm 0.326$ | $0.512 \pm 0.014$ | $0.435 \pm 0.123$ | $0.468 \pm 0.180$ | $1.0 \pm 0.0$ | $20 \pm 0$ |
| SWM | CF$-k$ | 1e-04 | $0.843 \pm 0.112$ | $0.522 \pm 0.017$ | $0.612 \pm 0.021$ | $0.698 \pm 0.003$ | $3.2 \pm 4.1$ | $19 \pm 0$ |
| | CF$-k$ | 1e-03 | $0.817 \pm 0.084$ | $0.539 \pm 0.019$ | $0.617 \pm 0.009$ | $0.707 \pm 0.005$ | $3.2 \pm 4.1$ | $19 \pm 0$ |
| | CF$-k$ | 1e-02 | $0.452 \pm 0.319$ | $0.503 \pm 0.013$ | $0.435 \pm 0.123$ | $0.468 \pm 0.180$ | $3.3 \pm 4.1$ | $19 \pm 0$ |
| | EU$-k$ | 1e-04 | $0.618 \pm 0.110$ | $0.537 \pm 0.015$ | $0.616 \pm 0.021$ | $0.696 \pm 0.002$ | $42.9 \pm 0.2$ | $19 \pm 0$ |
| | EU$-k$ | 1e-03 | $0.571 \pm 0.083$ | $0.514 \pm 0.019$ | $0.627 \pm 0.015$ | $0.710 \pm 0.005$ | $52.0 \pm 16.0$ | $19 \pm 0$ |
| | EU$-k$ | 1e-02 | $0.448 \pm 0.180$ | $0.501 \pm 0.016$ | $0.544 \pm 0.140$ | $0.577 \pm 0.157$ | $80.9 \pm 65.3$ | $19 \pm 0$ |
| | SalUn | 1e-05 | $0.796 \pm 0.089$ | $0.542 \pm 0.025$ | $0.630 \pm 0.003$ | $0.710 \pm 0.002$ | $1.3 \pm 0.0$ | $20 \pm 0$ |
| | SalUn | 1e-04 | $0.839 \pm 0.101$ | $0.536 \pm 0.023$ | $0.609 \pm 0.015$ | $0.701 \pm 0.009$ | $1.3 \pm 0.0$ | $21 \pm 0$ |
| | SalUn | 1e-03 | $0.839 \pm 0.090$ | $0.521 \pm 0.036$ | $0.606 \pm 0.014$ | $0.696 \pm 0.015$ | $1.3 \pm 0.0$ | $20 \pm 0$ |
| DIS | BT | 1e-04 | $0.845 \pm 0.096$ | $0.531 \pm 0.009$ | $0.623 \pm 0.004$ | $0.697 \pm 0.011$ | $2.1 \pm 0.0$ | $20 \pm 0$ |
| | BT | 1e-03 | $0.866 \pm 0.011$ | $0.529 \pm 0.041$ | $0.570 \pm 0.045$ | $0.650 \pm 0.048$ | $2.2 \pm 0.1$ | $20 \pm 0$ |
| | BT | 1e-02 | $0.803 \pm 0.062$ | $0.515 \pm 0.019$ | $0.550 \pm 0.019$ | $0.604 \pm 0.034$ | $2.1 \pm 0.0$ | $20 \pm 0$ |
| | SCRUB | 1e-04 | $0.857 \pm 0.126$ | $0.523 \pm 0.013$ | $0.612 \pm 0.021$ | $0.698 \pm 0.003$ | $1.4 \pm 0.0$ | $20 \pm 0$ |
| | SCRUB | 1e-03 | $0.831 \pm 0.108$ | $0.506 \pm 0.011$ | $0.617 \pm 0.009$ | $0.707 \pm 0.005$ | $1.5 \pm 0.0$ | $20 \pm 0$ |
| | SCRUB | 1e-02 | $0.443 \pm 0.304$ | $0.495 \pm 0.006$ | $0.435 \pm 0.123$ | $0.468 \pm 0.180$ | $1.4 \pm 0.0$ | $20 \pm 0$ |
| WI | FF | 1e-08 | $0.712 \pm 0.096$ | $0.508 \pm 0.013$ | $0.510 \pm 0.049$ | $0.614 \pm 0.039$ | $6.8 \pm 0.1$ | $20 \pm 0$ |
| | FF | 1e-07 | $0.718 \pm 0.171$ | $0.514 \pm 0.008$ | $0.501 \pm 0.088$ | $0.592 \pm 0.027$ | $6.7 \pm 0.0$ | $20 \pm 0$ |
| | FF | 1e-06 | $0.830 \pm 0.108$ | $0.526 \pm 0.017$ | $0.607 \pm 0.020$ | $0.683 \pm 0.013$ | $6.9 \pm 0.1$ | $20 \pm 0$ |
| | SSD | 1e-04 | $0.823 \pm 0.099$ | $0.527 \pm 0.009$ | $0.630 \pm 0.004$ | $0.711 \pm 0.002$ | $1.2 \pm 0.0$ | $20 \pm 0$ |
| | SSD | 1e-03 | $0.810 \pm 0.118$ | $0.519 \pm 0.024$ | $0.630 \pm 0.004$ | $0.711 \pm 0.002$ | $1.2 \pm 0.0$ | $20 \pm 0$ |
| | SSD | 1e-02 | $0.803 \pm 0.084$ | $0.522 \pm 0.031$ | $0.630 \pm 0.004$ | $0.711 \pm 0.002$ | $1.2 \pm 0.0$ | $20 \pm 0$ |

Table 23: Comparison of Unlearners on $GCN_{small}$ trained on BBBP (mean $\pm$ std).

| Group | Method | LR | LUMA | UMIA | F1 (test) | F1 (forget) | RunTime | GPU (MB) |
|---|---|---|---|---|---|---|---|---|
| − | Orig. | - | $0.618 \pm 0.010$ | $0.497 \pm 0.004$ | $0.665 \pm 0.014$ | $0.676 \pm 0.013$ | $93.3 \pm 0.6$ | $18 \pm 0$ |
| − | Gold | - | $0.636 \pm 0.000$ | $0.494 \pm 0.003$ | $0.681 \pm 0.012$ | $0.667 \pm 0.004$ | $93.3 \pm 0.6$ | $18 \pm 0$ |
| FT | GD | 1e-04 | $0.940 \pm 0.023$ | $0.496 \pm 0.002$ | $0.671 \pm 0.010$ | $0.678 \pm 0.014$ | $1.7 \pm 0.0$ | $18 \pm 0$ |
| | GD | 1e-03 | $0.935 \pm 0.024$ | $0.498 \pm 0.003$ | $0.682 \pm 0.009$ | $0.687 \pm 0.021$ | $1.7 \pm 0.0$ | $18 \pm 0$ |
| | GD | 1e-02 | $0.929 \pm 0.016$ | $0.495 \pm 0.004$ | $0.674 \pm 0.024$ | $0.673 \pm 0.027$ | $1.7 \pm 0.0$ | $18 \pm 0$ |
| | SRL | 1e-04 | $0.914 \pm 0.026$ | $0.500 \pm 0.009$ | $0.684 \pm 0.010$ | $0.694 \pm 0.019$ | $2.4 \pm 0.0$ | $18 \pm 0$ |
| | SRL | 1e-03 | $0.935 \pm 0.012$ | $0.496 \pm 0.005$ | $0.671 \pm 0.015$ | $0.669 \pm 0.015$ | $2.4 \pm 0.1$ | $18 \pm 0$ |
| | SRL | 1e-02 | $0.584 \pm 0.210$ | $0.492 \pm 0.013$ | $0.538 \pm 0.091$ | $0.526 \pm 0.075$ | $2.2 \pm 0.0$ | $18 \pm 0$ |
| | NG | 1e-04 | $0.724 \pm 0.191$ | $0.499 \pm 0.002$ | $0.566 \pm 0.083$ | $0.602 \pm 0.069$ | $0.5 \pm 0.0$ | $18 \pm 0$ |
| | NG | 1e-03 | $0.336 \pm 0.015$ | $0.494 \pm 0.005$ | $0.435 \pm 0.000$ | $0.432 \pm 0.000$ | $0.6 \pm 0.2$ | $18 \pm 0$ |
| | NG | 1e-02 | $0.335 \pm 0.016$ | $0.491 \pm 0.010$ | $0.435 \pm 0.000$ | $0.432 \pm 0.000$ | $0.6 \pm 0.2$ | $18 \pm 0$ |
| | ANG | 1e-04 | $0.912 \pm 0.029$ | $0.498 \pm 0.007$ | $0.652 \pm 0.028$ | $0.669 \pm 0.014$ | $1.3 \pm 0.1$ | $18 \pm 0$ |
| | ANG | 1e-03 | $0.480 \pm 0.092$ | $0.491 \pm 0.002$ | $0.484 \pm 0.045$ | $0.505 \pm 0.043$ | $1.5 \pm 0.2$ | $18 \pm 0$ |
| | ANG | 1e-02 | $0.335 \pm 0.015$ | $0.494 \pm 0.003$ | $0.435 \pm 0.000$ | $0.432 \pm 0.000$ | $1.4 \pm 0.2$ | $18 \pm 0$ |
| | UNSIR | 1e-04 | $0.938 \pm 0.019$ | $0.493 \pm 0.007$ | $0.671 \pm 0.010$ | $0.678 \pm 0.014$ | $2.1 \pm 0.0$ | $18 \pm 0$ |
| | UNSIR | 1e-03 | $0.925 \pm 0.029$ | $0.501 \pm 0.009$ | $0.682 \pm 0.009$ | $0.687 \pm 0.021$ | $2.1 \pm 0.0$ | $18 \pm 0$ |
| | UNSIR | 1e-02 | $0.926 \pm 0.021$ | $0.490 \pm 0.003$ | $0.674 \pm 0.024$ | $0.673 \pm 0.027$ | $2.1 \pm 0.0$ | $18 \pm 0$ |
| SWM | CF$-k$ | 1e-04 | $0.939 \pm 0.020$ | $0.491 \pm 0.004$ | $0.671 \pm 0.010$ | $0.678 \pm 0.014$ | $1.9 \pm 0.1$ | $18 \pm 0$ |
| | CF$-k$ | 1e-03 | $0.936 \pm 0.022$ | $0.497 \pm 0.001$ | $0.682 \pm 0.009$ | $0.687 \pm 0.021$ | $1.9 \pm 0.0$ | $18 \pm 0$ |
| | CF$-k$ | 1e-02 | $0.920 \pm 0.020$ | $0.491 \pm 0.008$ | $0.674 \pm 0.024$ | $0.673 \pm 0.027$ | $1.9 \pm 0.0$ | $18 \pm 0$ |
| | EU$-k$ | 1e-04 | $0.691 \pm 0.048$ | $0.491 \pm 0.004$ | $0.676 \pm 0.016$ | $0.678 \pm 0.014$ | $66.6 \pm 18.4$ | $18 \pm 0$ |
| | EU$-k$ | 1e-03 | $0.700 \pm 0.008$ | $0.492 \pm 0.006$ | $0.698 \pm 0.004$ | $0.692 \pm 0.013$ | $56.6 \pm 0.8$ | $18 \pm 0$ |
| | EU$-k$ | 1e-02 | $0.703 \pm 0.011$ | $0.502 \pm 0.008$ | $0.692 \pm 0.003$ | $0.689 \pm 0.008$ | $56.2 \pm 0.3$ | $18 \pm 0$ |
| | SalUn | 1e-04 | $0.918 \pm 0.019$ | $0.496 \pm 0.003$ | $0.684 \pm 0.010$ | $0.696 \pm 0.016$ | $2.7 \pm 0.1$ | $19 \pm 0$ |
| | SalUn | 1e-03 | $0.934 \pm 0.009$ | $0.495 \pm 0.004$ | $0.670 \pm 0.014$ | $0.669 \pm 0.015$ | $2.7 \pm 0.1$ | $19 \pm 0$ |
| | SalUn | 1e-02 | $0.556 \pm 0.203$ | $0.490 \pm 0.004$ | $0.535 \pm 0.088$ | $0.507 \pm 0.080$ | $2.8 \pm 0.1$ | $19 \pm 0$ |
| DIS | BT | 1e-04 | $0.820 \pm 0.065$ | $0.513 \pm 0.023$ | $0.615 \pm 0.005$ | $0.693 \pm 0.020$ | $4.2 \pm 0.1$ | $19 \pm 0$ |
| | BT | 1e-03 | $0.731 \pm 0.190$ | $0.521 \pm 0.011$ | $0.612 \pm 0.091$ | $0.653 \pm 0.103$ | $4.2 \pm 0.0$ | $18 \pm 0$ |
| | BT | 1e-02 | $0.650 \pm 0.297$ | $0.504 \pm 0.010$ | $0.554 \pm 0.114$ | $0.561 \pm 0.118$ | $4.3 \pm 0.0$ | $18 \pm 0$ |
| | SCRUB | 1e-04 | $0.929 \pm 0.021$ | $0.493 \pm 0.005$ | $0.671 \pm 0.010$ | $0.678 \pm 0.014$ | $3.1 \pm 0.0$ | $19 \pm 0$ |
| | SCRUB | 1e-03 | $0.923 \pm 0.028$ | $0.501 \pm 0.006$ | $0.682 \pm 0.009$ | $0.687 \pm 0.021$ | $3.1 \pm 0.0$ | $19 \pm 0$ |
| | SCRUB | 1e-02 | $0.924 \pm 0.018$ | $0.495 \pm 0.003$ | $0.674 \pm 0.024$ | $0.673 \pm 0.027$ | $3.1 \pm 0.0$ | $19 \pm 0$ |
| WI | FF | 1e-08 | $0.885 \pm 0.014$ | $0.498 \pm 0.008$ | $0.681 \pm 0.015$ | $0.692 \pm 0.012$ | $10.3 \pm 0.1$ | $19 \pm 0$ |
| | FF | 1e-07 | $0.668 \pm 0.268$ | $0.494 \pm 0.007$ | $0.576 \pm 0.122$ | $0.589 \pm 0.111$ | $10.2 \pm 0.1$ | $19 \pm 0$ |
| | FF | 1e-06 | $0.882 \pm 0.027$ | $0.493 \pm 0.007$ | $0.661 \pm 0.028$ | $0.661 \pm 0.016$ | $10.2 \pm 0.2$ | $19 \pm 0$ |
| | SSD | 1e-04 | $0.931 \pm 0.026$ | $0.496 \pm 0.003$ | $0.665 \pm 0.014$ | $0.676 \pm 0.013$ | $2.5 \pm 0.1$ | $19 \pm 0$ |
| | SSD | 1e-03 | $0.931 \pm 0.025$ | $0.494 \pm 0.004$ | $0.665 \pm 0.014$ | $0.676 \pm 0.013$ | $2.5 \pm 0.0$ | $19 \pm 0$ |
| | SSD | 1e-02 | $0.927 \pm 0.031$ | $0.496 \pm 0.006$ | $0.665 \pm 0.014$ | $0.676 \pm 0.013$ | $2.6 \pm 0.0$ | $19 \pm 0$ |

Table 24: Comparison of Unlearners on $GCN_{large}$ trained on BBBP (mean $\pm$ std).

| Group | Method | LR | LUMA | UMIA | F1 (test) | F1 (forget) | RunTime | GPU (MB) |
|-------|--------|-----|------|------|-----------|-------------|---------|----------|
| – | Orig. | - | $0.617_{\pm 0.002}$ | $0.501_{\pm 0.007}$ | $0.702_{\pm 0.003}$ | $0.705_{\pm 0.007}$ | $55.1_{\pm 0.2}$ | $18_{\pm 0}$ |
| – | Gold | - | $0.636_{\pm 0.000}$ | $0.498_{\pm 0.005}$ | $0.712_{\pm 0.007}$ | $0.689_{\pm 0.007}$ | $55.1_{\pm 0.2}$ | $18_{\pm 0}$ |
| FT | GD | 1e-04 | $0.924_{\pm 0.014}$ | $0.499_{\pm 0.006}$ | $0.705_{\pm 0.005}$ | $0.709_{\pm 0.004}$ | $1.0_{\pm 0.0}$ | $19_{\pm 0}$ |
| | GD | 1e-03 | $0.921_{\pm 0.034}$ | $0.493_{\pm 0.006}$ | $0.717_{\pm 0.008}$ | $0.715_{\pm 0.022}$ | $1.0_{\pm 0.0}$ | $19_{\pm 0}$ |
| | GD | 1e-02 | $0.872_{\pm 0.110}$ | $0.496_{\pm 0.004}$ | $0.668_{\pm 0.044}$ | $0.657_{\pm 0.030}$ | $1.0_{\pm 0.0}$ | $18_{\pm 0}$ |
| | SRL | 1e-04 | $0.930_{\pm 0.013}$ | $0.496_{\pm 0.004}$ | $0.710_{\pm 0.005}$ | $0.711_{\pm 0.006}$ | $1.4_{\pm 0.0}$ | $19_{\pm 0}$ |
| | SRL | 1e-03 | $0.733_{\pm 0.309}$ | $0.497_{\pm 0.010}$ | $0.628_{\pm 0.134}$ | $0.608_{\pm 0.110}$ | $1.5_{\pm 0.0}$ | $19_{\pm 0}$ |
| | SRL | 1e-02 | $0.777_{\pm 0.127}$ | $0.507_{\pm 0.015}$ | $0.633_{\pm 0.039}$ | $0.630_{\pm 0.044}$ | $1.3_{\pm 0.0}$ | $19_{\pm 0}$ |
| | NG | 1e-04 | $0.408_{\pm 0.126}$ | $0.495_{\pm 0.001}$ | $0.485_{\pm 0.046}$ | $0.493_{\pm 0.059}$ | $0.3_{\pm 0.0}$ | $19_{\pm 0}$ |
| | NG | 1e-03 | $0.285_{\pm 0.007}$ | $0.498_{\pm 0.007}$ | $0.435_{\pm 0.000}$ | $0.432_{\pm 0.000}$ | $0.3_{\pm 0.0}$ | $19_{\pm 0}$ |
| | NG | 1e-02 | $0.284_{\pm 0.007}$ | $0.491_{\pm 0.001}$ | $0.435_{\pm 0.000}$ | $0.432_{\pm 0.000}$ | $0.3_{\pm 0.0}$ | $19_{\pm 0}$ |
| | ANG | 1e-04 | $0.916_{\pm 0.022}$ | $0.498_{\pm 0.011}$ | $0.688_{\pm 0.012}$ | $0.694_{\pm 0.009}$ | $0.9_{\pm 0.0}$ | $19_{\pm 0}$ |
| | ANG | 1e-03 | $0.311_{\pm 0.031}$ | $0.494_{\pm 0.002}$ | $0.441_{\pm 0.006}$ | $0.453_{\pm 0.020}$ | $0.9_{\pm 0.0}$ | $19_{\pm 0}$ |
| | ANG | 1e-02 | $0.284_{\pm 0.007}$ | $0.496_{\pm 0.003}$ | $0.435_{\pm 0.000}$ | $0.432_{\pm 0.000}$ | $0.9_{\pm 0.0}$ | $19_{\pm 0}$ |
| | UNSIR | 1e-04 | $0.928_{\pm 0.009}$ | $0.494_{\pm 0.006}$ | $0.705_{\pm 0.005}$ | $0.709_{\pm 0.004}$ | $1.3_{\pm 0.0}$ | $20_{\pm 0}$ |
| | UNSIR | 1e-03 | $0.917_{\pm 0.039}$ | $0.504_{\pm 0.002}$ | $0.717_{\pm 0.008}$ | $0.715_{\pm 0.022}$ | $1.4_{\pm 0.0}$ | $20_{\pm 0}$ |
| | UNSIR | 1e-02 | $0.866_{\pm 0.111}$ | $0.497_{\pm 0.007}$ | $0.668_{\pm 0.044}$ | $0.657_{\pm 0.030}$ | $1.3_{\pm 0.0}$ | $19_{\pm 0}$ |
| SWM | CF$-k$ | 1e-04 | $0.920_{\pm 0.008}$ | $0.488_{\pm 0.002}$ | $0.705_{\pm 0.005}$ | $0.709_{\pm 0.004}$ | $1.1_{\pm 0.0}$ | $19_{\pm 0}$ |
| | CF$-k$ | 1e-03 | $0.917_{\pm 0.034}$ | $0.490_{\pm 0.004}$ | $0.717_{\pm 0.008}$ | $0.715_{\pm 0.022}$ | $1.1_{\pm 0.0}$ | $19_{\pm 0}$ |
| | CF$-k$ | 1e-02 | $0.863_{\pm 0.110}$ | $0.491_{\pm 0.003}$ | $0.668_{\pm 0.044}$ | $0.657_{\pm 0.030}$ | $1.2_{\pm 0.0}$ | $19_{\pm 0}$ |
| | EU$-k$ | 1e-04 | $0.713_{\pm 0.005}$ | $0.492_{\pm 0.006}$ | $0.709_{\pm 0.001}$ | $0.705_{\pm 0.007}$ | $34.0_{\pm 0.0}$ | $19_{\pm 0}$ |
| | EU$-k$ | 1e-03 | $0.699_{\pm 0.021}$ | $0.492_{\pm 0.005}$ | $0.731_{\pm 0.004}$ | $0.712_{\pm 0.013}$ | $34.1_{\pm 0.1}$ | $19_{\pm 0}$ |
| | EU$-k$ | 1e-02 | $0.704_{\pm 0.011}$ | $0.492_{\pm 0.005}$ | $0.681_{\pm 0.008}$ | $0.693_{\pm 0.012}$ | $34.3_{\pm 0.1}$ | $19_{\pm 0}$ |
| | SalUn | 1e-04 | $0.919_{\pm 0.012}$ | $0.497_{\pm 0.006}$ | $0.711_{\pm 0.007}$ | $0.713_{\pm 0.002}$ | $1.7_{\pm 0.1}$ | $21_{\pm 0}$ |
| | SalUn | 1e-03 | $0.740_{\pm 0.328}$ | $0.504_{\pm 0.013}$ | $0.624_{\pm 0.134}$ | $0.608_{\pm 0.116}$ | $1.6_{\pm 0.1}$ | $20_{\pm 0}$ |
| | SalUn | 1e-02 | $0.689_{\pm 0.131}$ | $0.499_{\pm 0.005}$ | $0.601_{\pm 0.038}$ | $0.595_{\pm 0.055}$ | $1.7_{\pm 0.0}$ | $18_{\pm 0}$ |
| DIS | BT | 1e-04 | $0.879_{\pm 0.028}$ | $0.494_{\pm 0.008}$ | $0.659_{\pm 0.010}$ | $0.696_{\pm 0.007}$ | $2.9_{\pm 0.0}$ | $20_{\pm 0}$ |
| | BT | 1e-03 | $0.854_{\pm 0.043}$ | $0.497_{\pm 0.001}$ | $0.652_{\pm 0.030}$ | $0.686_{\pm 0.029}$ | $2.8_{\pm 0.0}$ | $20_{\pm 0}$ |
| | BT | 1e-02 | $0.438_{\pm 0.131}$ | $0.496_{\pm 0.002}$ | $0.510_{\pm 0.069}$ | $0.495_{\pm 0.058}$ | $3.0_{\pm 0.0}$ | $20_{\pm 0}$ |
| | SCRUB | 1e-04 | $0.917_{\pm 0.010}$ | $0.495_{\pm 0.014}$ | $0.705_{\pm 0.005}$ | $0.709_{\pm 0.004}$ | $1.9_{\pm 0.0}$ | $20_{\pm 0}$ |
| | SCRUB | 1e-03 | $0.920_{\pm 0.032}$ | $0.498_{\pm 0.004}$ | $0.717_{\pm 0.008}$ | $0.715_{\pm 0.022}$ | $1.9_{\pm 0.0}$ | $20_{\pm 0}$ |
| | SCRUB | 1e-02 | $0.864_{\pm 0.113}$ | $0.495_{\pm 0.007}$ | $0.668_{\pm 0.044}$ | $0.657_{\pm 0.030}$ | $1.9_{\pm 0.0}$ | $20_{\pm 0}$ |
| WI | FF | 1e-08 | $0.771_{\pm 0.167}$ | $0.499_{\pm 0.006}$ | $0.639_{\pm 0.064}$ | $0.657_{\pm 0.073}$ | $7.1_{\pm 0.1}$ | $20_{\pm 0}$ |
| | FF | 1e-07 | $0.787_{\pm 0.075}$ | $0.492_{\pm 0.002}$ | $0.652_{\pm 0.043}$ | $0.661_{\pm 0.049}$ | $7.1_{\pm 0.0}$ | $20_{\pm 0}$ |
| | FF | 1e-06 | $0.793_{\pm 0.147}$ | $0.505_{\pm 0.008}$ | $0.669_{\pm 0.066}$ | $0.669_{\pm 0.076}$ | $7.0_{\pm 0.0}$ | $20_{\pm 0}$ |
| | SSD | 1e-04 | $0.926_{\pm 0.004}$ | $0.492_{\pm 0.005}$ | $0.702_{\pm 0.003}$ | $0.705_{\pm 0.007}$ | $1.5_{\pm 0.1}$ | $20_{\pm 0}$ |
| | SSD | 1e-03 | $0.930_{\pm 0.006}$ | $0.495_{\pm 0.001}$ | $0.702_{\pm 0.003}$ | $0.705_{\pm 0.007}$ | $1.6_{\pm 0.1}$ | $20_{\pm 0}$ |
| | SSD | 1e-02 | $0.935_{\pm 0.006}$ | $0.497_{\pm 0.003}$ | $0.702_{\pm 0.003}$ | $0.705_{\pm 0.007}$ | $1.6_{\pm 0.1}$ | $20_{\pm 0}$ |

