# OpenReview forum: "A Multi-domain Benchmark for Machine Unlearning in Classification Tasks"
_ICLR.cc/2026/Conference — ICLR 2026 Conference Withdrawn Submission_

### Official Review · Reviewer_hDR4 · 2025-10-30

**Soundness:** 1
**Presentation:** 2
**Contribution:** 1
**Rating:** 2
**Confidence:** 5

**Summary:**

The paper proposes a multi-domain benchmark for machine unlearning, covering four modalities: image, text, tabular, and graph data, with two datasets per domain. The benchmark evaluates 12 existing unlearning methods. The authors introduce LUMA, a unified metric that combines utility, efficacy, and efficiency using a harmonic mean with Laplace kernel weighting, aiming to provide a balanced quantitative evaluation.

**Strengths:**

- The inclusion of image, text, tabular, and graph domains provides a broad empirical base for comparing unlearning algorithms beyond the typical image-only or text-only setting.
- Evaluating 12 well-known unlearning algorithms under a unified framework provides useful practical insights into their relative performance and trade-offs.
- Although simple, the effort to jointly quantify utility, efficacy, and efficiency highlights an important issue in unlearning evaluation—balancing these often conflicting objectives.
- The benchmark may lower entry barriers for researchers entering the unlearning field, potentially serving as a baseline for cross-domain comparison.

**Weaknesses:**

- The benchmark mainly aggregates existing datasets, models, and methods without introducing new algorithms or theoretical insights.
- CIFAR-100 and CelebA are legacy datasets; more recent benchmarks (ImageNet subsets, LAION, COCO, etc.) would be preferable.
- IMDB and AG News are outdated; newer benchmarks like TOFU or RWKU should be used.
- Relying on ResNet and BERT is no longer representative, recent unlearning works commonly use ViT or LLaMA-based backbones.
- The paper’s title implies class-level unlearning, but it is unclear whether the experiments actually target class-unlearning or instance-level unlearning. The distinction is important, as the forgetting objective and evaluation criteria differ substantially between the two.
- LUMA is a simple harmonic mean of three normalized metrics with Laplace kernels, lacking clear theoretical grounding or interpretability. Moreover, it is unclear why a unified metric like LUMA is even necessary, since utility, efficacy, and efficiency inherently capture different and sometimes orthogonal aspects of unlearning performance.

**Questions:**

see weaknesses

---

### Official Review · Reviewer_7daK · 2025-11-01

**Soundness:** 2
**Presentation:** 3
**Contribution:** 2
**Rating:** 2
**Confidence:** 4

**Summary:**

This paper presents a comprehensive MU benchmark across 4 modalities (image, text, tabular, graph), evaluating 12 methods on 8 datasets. The authors introduce LUMA, a unified metric combining utility, efficacy, and efficiency. While this represents the most extensive MU benchmark to date and fills important gaps in tabular and graph domains, several methodological concerns limit its practical impact.

**Strengths:**

1. This is the first comprehensive multi-domain benchmark that systematically covers previously neglected tabular and graph domains, which is a significant contribution to the field.
2. The proposed taxonomy organizing methods into 4 categories (Fine-Tuning, Selective Weight Modification, Distillation, and Weight Importance) provides a useful framework for understanding the MU landscape.

**Weaknesses:**

1. The paper directly applies general MU methods to specialized domains without acknowledging that domain-specific unlearning approaches exist. For instance, graph unlearning methods like GraphEraser are designed to handle structural dependencies that generic methods cannot address effectively, which limits the practical relevance of findings in these domains.
2. The choice of 20% as the forget set size lacks proper justification, and the claimed consistency with "prior MU literature" needs specific citations. In privacy-focused scenarios, forget sets are typically much smaller (single-digit percentages or individual samples), which significantly impacts the difficulty and practical relevance of the unlearning task.
3. Using a 1-layer GCN as one of the graph models is problematic since single-layer GNNs cannot effectively capture multi-hop neighborhood information fundamental to GNN performance. The literature widely recognizes that GNNs require at least 2-3 layers to demonstrate their advantages, which undermines the validity of graph domain conclusions.
4. The benchmark omits influence function-based unlearning methods, which represent a theoretically principled and practically important approach in the MU literature. This is a notable gap given the paper's claim of comprehensive coverage.
5. The efficiency measurement using peak GPU memory is misleading because it does not reflect actual computational cost. Methods like retraining may show lower peak memory but consume more resources over time due to longer execution, making this metric an inaccurate representation of true efficiency.
6. Despite LUMA's technical sophistication, reducing unlearning performance to a single score has fundamental limitations. Different application scenarios prioritize different objectives: privacy-critical applications demand maximum efficacy regardless of cost, while production systems may prioritize efficiency. A single aggregate score cannot capture these trade-offs adequately and may mislead practitioners about which method suits their specific needs.

**Questions:**

s use 20% forget sets to justify this choice? How do results change with more realistic proportions like 1%, 5%, or 10% that are common in privacy scenarios?
2. Can you justify using a 1-layer GCN given that such architectures cannot leverage GNN advantages? What is the performance gap between 1-layer and 2-layer GCNs on these datasets before unlearning?
3. Have you considered more meaningful efficiency metrics like memory-time product or energy consumption instead of peak memory? How would these affect method rankings?
4. Rather than a single LUMA score, have you considered presenting Pareto frontiers showing trade-offs between utility, efficacy, and efficiency? This would better serve practitioners with diverse requirements.

---

### Official Review · Reviewer_MarZ · 2025-11-01

**Soundness:** 2
**Presentation:** 3
**Contribution:** 1
**Rating:** 2
**Confidence:** 4

**Summary:**

The paper provides a comprehensive studies of the unlearning approximation algorithms on different data types based on the comprehensive evaluation of unlearning metrics. The paper introduces a Laplacian Unlearning Multidimensional Assessment,
a unified metric that consolidates them into a single score.

**Strengths:**

the comprehensive study of unlearning approximation algorithms on different domains and data types all in one place.

**Weaknesses:**

I appriciate the authors' hard work to investigate MU methods from different perspective and providing a compreheisve study on the different domains and models, but I think there are still so much room to work on the contribution of the paper for this research domain.


Line 36 // "Machine unlearning .... ": exact Machine unlearning is retraining from scratch -  Approximate machine unlearning is the response to address the retraining from scratch.

line 50 //""training appropriate model ... "": ambiguous, bad grammar.


line 76 // It is standard and common practice in the domain of MU that these aspects of model would be evaluated with different metrics.

**Questions:**

line 50 // "Applying unlearning methods ... " : Could you justify this point? Since,  many of the unlearning methods are domain independent and don't need to make any adjustments.

Line 100 // Understanding and evaluating the unlearning approximation method on different domains may not be investigated simultaneously in a single work, but the algorithms such as Gradient Ascent or NegGrad have been well investigated in many domains. from vision to graph neural networks [1,2]. So the question is what understanding we can achieve from evaluating them together in a single research?


line 106 // "the incorporation of ...": Why we want to incorporate the evaluation metrics of MU in one single metric? What would be the advantage of that? Don't we want to investigate and gain more knowledge about how a method perform from different aspect rather than providing a single score?



[1]Cheng, Jiali, et al. "Gnndelete: A general strategy for unlearning in graph neural networks." arXiv preprint arXiv:2302.13406 (2023).

[2]Jia, Jinghan, et al. "Model sparsity can simplify machine unlearning." Advances in Neural Information Processing Systems 36 (2023): 51584-51605.

---

### Official Review · Reviewer_jZd3 · 2025-11-01

**Soundness:** 2
**Presentation:** 2
**Contribution:** 2
**Rating:** 4
**Confidence:** 3

**Summary:**

The paper presents an extensive benchmark and survey of machine unlearning methods across multiple data modalities, including images, text, tabular, and graph data. It proposes a unified metric, LUMA, to jointly evaluate unlearning methods along the axes of utility, efficacy, and efficiency, and systematically compares representative unlearning techniques across eight datasets.

**Strengths:**

- The benchmark systematically evaluates unlearning algorithms not only on standard image datasets (e.g., CIFAR-100, CelebA) but also on tabular and  graph data
- The benchmark provides valuable takeaways on learning-rate sensitivity and model-size robustness across methods.
- The analysis highlights which unlearners are stable or sensitive across hyperparameter scales and datasets.

**Weaknesses:**

- By forcing a single evaluation protocol across all data types, the benchmark focuses only on methods that can operate on all domains, which excludes many strong domain-specific unlearners (e.g., GNN-specific or LLM-specific methods). This constraint, while necessary for consistency, inherently limits the scope of the conclusions.
- The benchmark omits some  recent unlearning methods, in particular DELETE (CVPR 2025)
- No  discussion is provided (neither in the related works) for LLM-based unlearning, despite recent literature such as “Rethinking Machine Unlearning for Large Language Models” (S. Liu et al., 2024). For graphs, only classification datasets are tested, and the paper omits graph-specific unlearning frameworks, such as:
Wu et al., Certified Edge Unlearning for Graph Neural Networks
Chen et al., Graph Unlearning
Said et al., A Survey of Graph Unlearning
- While innovative, LUMA may obscure specific aspects of performance. Practitioners often need to see utility (model accuracy) and running time separately rather than as an aggregated score.
- The figure visualizing learning-rate sensitivity (Figure 2) is dense and visually hard to parse. The takeaways described in the text (e.g., LR sensitivity and size invariance) are not immediately evident without extensive reading. Simplifying this visualization (e.g., separate panels per unlearner or clearer legends) would improve accessibility.

Minor:
- add requirements in the repo

**Questions:**

- In practice, how dominant are the efficiency components in the final LUMA score?
- What motivated setting $\gamma =3 $ for  $M_u$ and $M_e$ ?
- Similarly, how were the weighting coefficients $w_i$ chosen in LUMA?

---

### Note · Authors · 2025-12-03

**Comment:**

I am writing to request the withdrawal of my paper. I appreciate your assistance.

**Withdrawal Confirmation:**

I have read and agree with the venue's withdrawal policy on behalf of myself and my co-authors.